# The telomere binding protein Pot1 maintains haematopoietic stem cell activity with age

Kentaro Hosokawa[1,2], Ben D. MacArthur [3], Yoshiko Matsumoto Ikushima[2,4], Hirofumi Toyama[2], Yoshikazu Masuhiro[5], Shigemasa Hanazawa[5], Toshio Suda[2,6] & Fumio Arai [1,2]

Repeated cell divisions and aging impair stem cell function. However, the mechanisms by which this occurs are not fully understood. Here we show that protection of telomeres 1A (Pot1a), a component of the Shelterin complex that protects telomeres, improves haematopoietic stem cell (HSC) activity during aging. Pot1a is highly expressed in young HSCs, but declines with age. In mouse HSCs, Pot1a knockdown increases DNA damage response (DDR) and inhibits self-renewal. Conversely, Pot1a overexpression or treatment with POT1a protein prevents DDR, maintained self-renewal activity and rejuvenated aged HSCs upon ex vivo culture. Moreover, treatment of HSCs with exogenous Pot1a inhibits the production of reactive oxygen species, suggesting a non-telomeric role for Pot1a in HSC maintenance. Consistent with these results, treatment with exogenous human POT1 protein maintains human HSC activity in culture. Collectively, these results show that Pot1a/POT1 sustains HSC activity and can be used to expand HSC numbers ex vivo.

[1] Department of Stem Cell Biology and Medicine, Graduate School of Medical Sciences, Kyushu University, 3-1-1 Maidashi, Higashi-ku, Fukuoka 812-8582, Japan. [2] Department of Cell Differentiation, The Sakaguchi Laboratory of Developmental Biology, School of Medicine, Keio University, 35 Shinanomachi, Shinjuku-ku, Tokyo 160-8582, Japan. [3] Centre for Human Development, Stem Cells and Regeneration, Faculty of Medicine, University of Southampton, University Road, Southampton SO17 1BJ, UK. [4] National Center for Global Health and Medicine, 1-21-1 Toyama, Shinjuku-ku, Tokyo 162-8655, Japan. [5] Laboratory of Molecular and Cellular Physiology, Department of Applied Biological Sciences, College of Bioresource Sciences, Nihon University, Fujisawa City, Kanagawa 252-0880, Japan. [6] Cancer Science Institute of Singapore, National University of Singapore, Singapore 117599, Singapore. Correspondence and requests for materials should be addressed to T.S. (email: sudato@keio.jp) or to F.A. (email: farai@scr.med.kyushu-u.ac.jp)

Appropriate regulation of haematopoietic stem cell (HSC) self-renewal is critical for the maintenance of life long hematopoiesis. However, long-term repeated cell divisions induce the accumulation of DNA damage, which, along with replication stress, significantly compromises HSC function[1–6]. This sensitivity to stress-induced DNA-damage is a primary obstacle to establishing robust protocols for the ex vivo expansion of functional HSCs. Telomeres are particularly sensitive to such damage because they are fragile sites in the genome[3, 7, 8]. As HSCs lose telomeric DNA with each cell division[9], which ultimately limits their replicative potential[10], HSCs therefore require a protective mechanism to prevent DNA damage response (DDR) at telomeres in order to maintain their function.

The shelterin complex—which contains six subunit proteins, TRF1, TRF2, POT1, TIN2, TPP1, and RAP1—has a crucial role in the regulation of telomere length and loop structure, as well as in the protection of telomeres from ataxia telangiectasia-mutated (ATM) and ATM- and RAD3-related (ATR) dependent DDR signaling pathways[11, 12]. Protection of telomeres 1 (POT1) binds to telomeric single-stranded DNA (ssDNA) through its oligonucleotide/oligosaccharide-binding fold domains (OB domains)[13, 14] and thereby prevents ATR signaling by blocking replication protein A (RPA), the ssDNA binding protein that activates the ATR pathway[15]. Furthermore, POT1 can also bind to sub-telomeric and non-telomeric DNA through its OB1 domain, which recognizes an OB1-biding motif (TTAGG) and a non-telomeric motif, suggesting further non-telomeric functions for POT1 related to gene transcription, replication, or repair[16].

Human shelterin contains a single POT1 protein, whereas the mouse genome has two POT1 orthologs, Pot1a and Pot1b, which have different functions at telomeres[17]. Pot1a is required for the repression of DDR at telomeres[17, 18]. In contrast, Pot1b is involved in the maintenance of telomere terminus structure[17, 19, 20]. Commensurate with these different roles, Pot1a knockout (KO) mice have early embryonic lethality, whereas Pot1b KO mice remain alive and fertile and exhibit a dyskeratosis congenita-like phenotype when generated in a telomerase-haploinsufficient background[17, 20]. It has recently been shown that shelterin components, TRF1, Pot1b, and Tpp1, critically regulate HSC activity and survival[21–23]. However, due to embryonic lethality, the role of Pot1a in maintaining HSC function is still unclear and it is not known if POT1/Pot1a has a non-telomeric role in HSC regulation and maintenance.

Here, we show that Pot1a maintains HSC activity by protecting against DNA damage and preventing the production of reactive oxygen spices (ROS). Due to these protective functions, we find that treatment with exogenous Pot1a maintains HSC self-renewal and function ex vivo and improves the activity of aged HSCs.

## Results

**Pot1a expression in HSCs.** First, we analyzed the expression of Pot1a in haematopoietic stem, progenitor and differentiated cells. We observed that Pot1a is expressed at substantially higher levels in short-term (ST)- and long-term (LT)-HSC fractions than in progenitor and differentiated cell fractions (Fig. 1a–d), yet this expression sharply decreases with age (Fig. 1e–g). Other components of the shelterin complex were also more highly expressed in HSC fractions than in progenitor and differentiated cell fractions (Supplementary Fig. 1a) and showed similar expression changes with aging, with the exception of Terf1 and Rap1 (Supplementary Fig. 1b). These data indicate a close correspondence between Pot1a expression and aging in LT-HSCs.

**Loss of Pot1a compromises LT-HSC activity.** To investigate the function of Pot1a in the regulation of HSC maintenance,

we transduced retrovirus-expressing Pot1a-specific shRNAs (shPot1a-1 and -2) into lineage−Sca-1+c-Kit+ (LSK) CD41−CD48−CD150+ cells. Both Pot1a shRNAs substantially suppressed Pot1a expression at both the messenger RNA (mRNA) and protein levels (Supplementary Fig. 2a, b). We then analyzed the effect of Pot1a knockdown on the function of HSCs. Knockdown of Pot1a decreased both the number of colony forming units in culture (CFU-Cs) and high-proliferative potential colony forming cells (HPP-CFCs) (Supplementary Fig. 2c, d). Furthermore, while transduction of shPot1a increased total cell numbers it also inhibited LT-HSC proliferation and induced apoptosis of LT-HSCs in culture (Supplementary Fig. 2e–g), suggesting that loss of Pot1a compromises HSC activity in vitro. Knockdown of Pot1a also markedly reduced donor cell engraftment after BM transplantation (BMT) in peripheral blood (PB), LSK and LT-HSC fractions (Supplementary Fig. 2h, j), and induced a bias toward myeloid and away from lymphoid differentiation (Supplementary Fig. 2i). Similar results have been previously observed with Tpp1 deficiency[21]. As TPP1 is required for the binding of POT1 to telomeres[24], this suggests that the function of Pot1a at telomeres is critical for HSC maintenance. To confirm the specificity of these results we co-transduced shPot1a and a silent mutant Pot1a (smPot1a), which has silent mutations within the shRNA targeting sequences, into LT-HSCs and transplanted them into lethally irradiated recipient mice. Expression of Pot1a was restored in co-transduced cells, and they accordingly showed substantially better engraftment than those that were transduced with shPot1a-1 or -2 alone (Supplementary Fig. 2k), indicating that expression of smPot1a was able to rescue the BM reconstitution activity of Pot1a knockdown HSCs.

To investigate these results further we then analyzed the effect of Pot1a knockdown on HSC gene expression patterns. First, we checked the expression levels of other shelterin components after Pot1a knockdown and found that the knockdown of Pot1a decreased the expression of Pot1b and Tpp1 but did not affect the expression levels of other shelterin components (Supplementary Fig. 3a). We then examined the expression of genes related to HSC aging, maintenance, apoptosis, and differentiation. Analysis of gene expression patterns revealed that Pot1a knockdown upregulated expression of genes associated with senescence/apoptosis and differentiation, and downregulated expression of genes required for HSC maintenance (Supplementary Fig. 3b). Collectively, these data suggest that Pot1a has an important role in regulating the proliferation, differentiation, and maintenance of LT-HSCs both in vitro and in vivo.

**Overexpression of Pot1a enhances HSC function.** To dissect Pot1a function further, we transduced a retrovirus-expressing Pot1a into LSK cells to assess the effect of Pot1a overexpression on stem cell function. We first checked whether Pot1a overexpression affects the level of other shelterin components and we found that the transduction of Pot1a did not change the expression level of other shelterin genes (Supplementary Fig. 4a). Next, we checked the effect of Pot1a overexpression on the function of LT-HSCs. As expected, overexpression of Pot1a substantially increased the numbers of CFU-Cs and HPP-CFCs, and colony sizes in culture (Fig. 2a–e). In accordance with this enhanced colony forming ability, expression of HSC markers was also upregulated (Supplementary Fig. 4b). To assess whether overexpression of Pot1a in culture affects self-renewal of HSCs in vivo, LSK cells overexpressing Pot1a were cultured for 7 or 14 days and then transplanted into lethally irradiated mice. Overexpression of Pot1a was observed to facilitate donor cell engraftment (Fig. 2f), indicating that in addition to maintaining HSC numbers, exogenous Pot1a also preserves the long-term

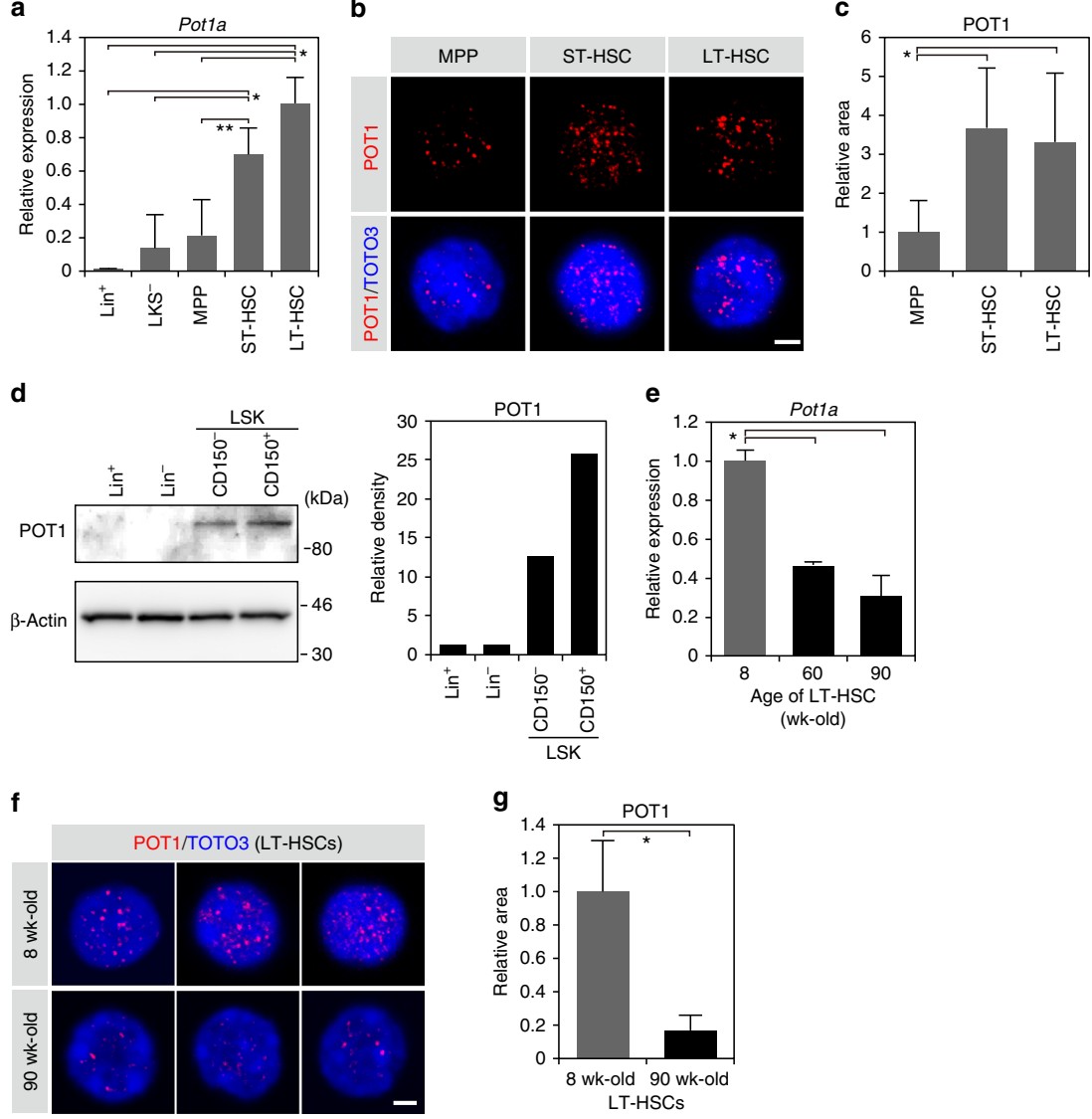

**Fig. 1** Expression of Pot1a in HSPCs. **a** Expression of *Pot1a* in: Lineage+ (Lin+) cells; Lin−Kit+Sca-1− (LKS−) cells; LSKCD41+CD48+CD150− multipotent progenitor (MPP) cells; LSKCD41+CD48+CD150+ cells (ST-HSCs); LSKCD41−CD48−CD150+ cells (LT-HSCs) isolated from 8 week-old mice. Data are expressed as the mean ± SD ($n = 4$, *$p < 0.01$, **$p < 0.05$ by Tukey's test). Representative data from two independent experiments are shown. **b** Immunocytochemical staining of POT1 (red) in HSPCs. Nucleus is stained by TOTO3 (blue). Scale bar, 2 μm. **c** Relative area of POT1 immunofluorescent dots in HSPCs. Fluorescence images were analyzed with ImageJ. Data are expressed as the mean ± SD ($n = 64$: MPP, $n = 67$: ST-HSCs, $n = 54$: LT-HSCs, *$p < 0.01$ by Tukey's test). Representative data from two independent experiments are shown. **d** Immunoblot analysis of POT1 in Lin+, Lin−, LSKCD150−, and LSKCD150+ cells (left panel). Densitometry analysis of the immunoblot is shown in the *right panel*. Representative data from two independent experiments are shown. **e** Expression of *Pot1a* in 8, 60, and 90 week-old LT-HSCs. Data are expressed as the mean ± SD ($n = 4$, *$p < 0.01$ by Tukey's test). Representative data from two independent experiments are shown. **f** Immunocytochemical staining of POT1 (red) in LT-HSCs isolated from 8 and 90 week-old mice. *Scale bar*, 2 μm. **g** Relative area of POT1 immunofluorescent dots in LT-HSCs isolated from 8 and 90 week-old mice. Data are expressed as the mean ± SD ($n = 146$: 8-week-old, $n = 180$: 90-week-old, *$p < 0.01$ by t-test)

reconstitution (LTR) ability of HSCs during in vitro culture. To investigate further, we examined the effect of Pot1a over-expression on maintenance of HSC self-renewal activity in serial BMT. Pot1a overexpression had no effect on donor cell engraftment in PB in primary BMT, although it did increase the proportion of donor-derived ST- and LT-HSC fraction compared with control GFP-transduced cells (Fig. 3a, Supplementary Fig. 5a, b). Upon secondary BMT we found that Pot1a overexpression substantially increased reconstitution in the PB, BM LSK, and LT-HSC fractions (Fig. 3c). Furthermore, Pot1a overexpression maintained a Ki67− quiescent population in the donor-derived LSK fraction after primary and secondary BMT

(Supplementary Fig. 5c). Notably, in contrast to Pot1a knockdown, Pot1a overexpression induced a distinct bias towards lymphoid differentiation and away from myeloid differentiation in donor cells (Fig. 3b, d). Taken together these data indicate that overexpression of Pot1a enhances the functional activity of HSCs in vivo.

**Pot1a regulates HSC self-renewal by preventing DDR.** To determine the mechanisms by which Pot1a regulates HSC activity we sought to assess the role of Pot1a in the protection of DDR signaling. Telomere dysfunction-induced foci (TIFs), a marker for DNA damage at telomeric DNA[25], were identified

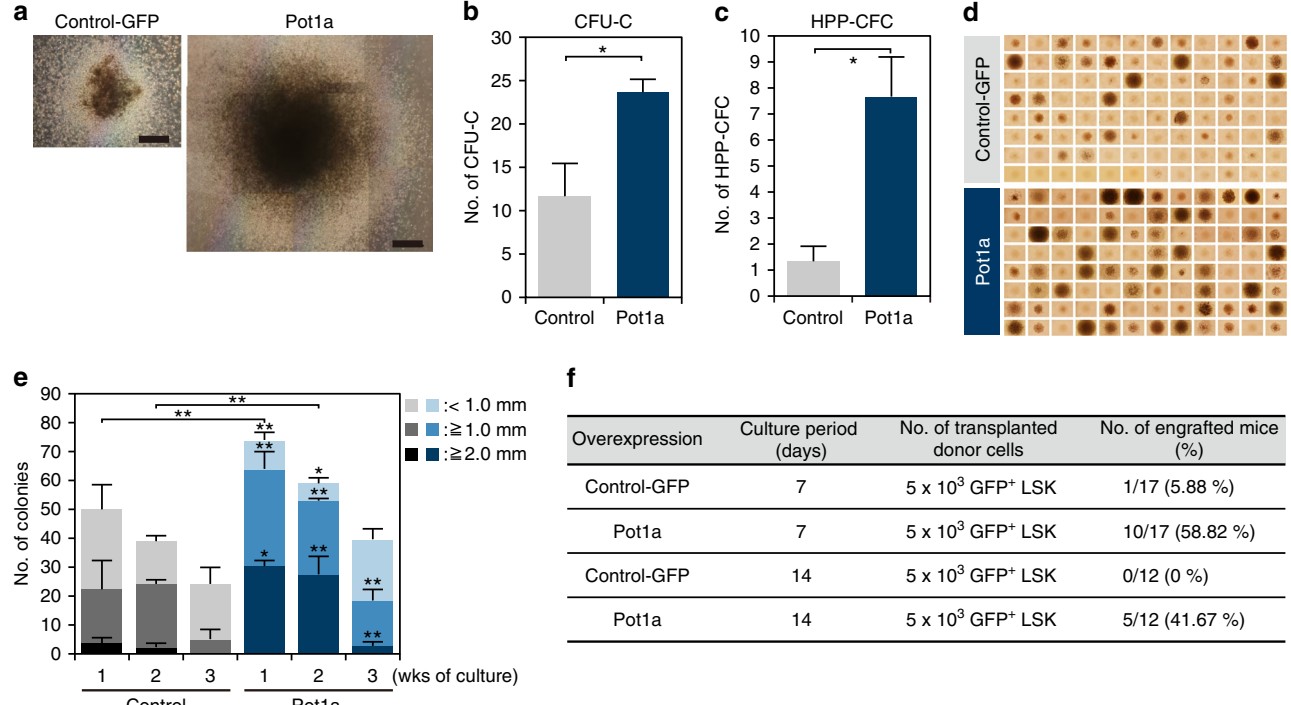

**Fig. 2** Overexpression of Pot1a enhances HSC proliferation activity. **a–c** Control GFP or Pot1a-transduced LSK cells isolated from 8 week-old mice were cultured for 2 weeks and their ability to form colonies was analyzed. **a** Representative colonies from control GFP (*left*) and Pot1a-transduced LSK cells (*right*) after 2 weeks in vitro culture. *Scale bar*, 0.5 mm. **b** Number of CFU-Cs. **c** Number of HPP-CFCs. Data are expressed as the mean ± SD ($n = 3$, *$p < 0.01$ by *t*-test). **d, e** Control GFP or Pot1a-transduced LSK cells were clone-sorted and cultured for 2 weeks in SF-O3 medium supplemented with SCF (20 ng ml$^{-1}$), TPO (50 ng ml$^{-1}$), IL-3 (20 ng ml$^{-1}$) and EPO (1 U ml$^{-1}$). **d** Development of colonies from single cells after 2 weeks in vitro culture. **e** Number of colonies derived from single cells after 1–3 weeks in vitro culture. Each colony was classified into three categories by their size (<1 mm, ≥1 mm, ≥2 mm). Data are expressed as the mean ± SD ($n = 3$, *$p < 0.01$, **$p < 0.05$ by Tukey's test). **f** Control GFP or Pot1a-transduced 8 week-old LSK cells were cultured for 7–14 days. Subsequently, $5 \times 10^3$ GFP$^+$LSK cells were transplanted into lethally irradiated mice. The number of recipient mice that showed long-term reconstitution of donor cells (>1% of GFP$^+$Ly5.1$^+$ cells in PB) is shown

by co-localization of 53BP1 and TRF1. Significant co-localization of 53BP1 and TRF1 was determined by image analysis (see Methods and Supplementary Table 1). We first observed that Pot1a knockdown increased the number of TIFs in LSK cells in culture (Fig. 4a, b). To determine whether the formation of TIFs in LT-HSCs was due to ATR signaling, we then measured the expression of replication protein A 32 kDa subunit (RPA32, a ssDNA binding protein that activates the ATR kinase pathway[8]) and phosphorylated Chk1 (pChk1, which has a critical role in DNA damage checkpoint control and tumor suppression[26, 27]) by immunocytochemistry, following BMT of control GFP or Pot1a overexpressing HSCs. We observed that HSCs overexpressing Pot1a had significantly fewer TIFs than controls 4 months after BMT (Fig. 4c, d) and expression of RPA32 (Fig. 4e) and pChk1 (Fig. 4f) were reduced upon Pot1a overexpression. Furthermore, Pot1a overexpression substantially reduced the phosphorylation of Chk1 after a second BMT (Fig. 4g). Interestingly, in addition to TIFs, we also observed that the total number of 53BP1 foci and the overall level of 53BP1 expression in LT-HSCs was increased by Pot1a knockdown and was decreased by Pot1a overexpression (Supplementary Fig. 6), suggesting that Pot1a protects both telomeric and extra-telomeric DNA in HSCs. We also found that overexpression of Pot1a increased telomerase activity in HSCs (Supplementary Fig. 4c), while Pot1a overexpression did not alter telomere length in donor-derived cells isolated 4 months after primary BMT (Supplementary Fig. 4d, e). These results indicate that the protection of both telomeric and non-telomeric DDR by Pot1a is important for the maintenance of HSC function.

**Pot1a prevents the production of ROS in LT-HSCs**. As we observed that Pot1a expression is negatively associated with 53BP1 levels, we sought to clarify how Pot1a regulates non-telomeric DDR in LT-HSCs. To do so, we performed microarray analysis of control-GFP and Pot1a-transduced LT-HSCs (Fig. 5a). Gene set enrichment analysis (GSEA) revealed that Pot1a transduction reduced the expression of genes associated with oxidative phosphorylation, mitochondrial respiratory chain, and oxygen and reactive oxygen species metabolic processes by comparison with controls (Fig. 5b). As oxidative stress induced by ROS is known to induce DNA damage[6] and reduce the self-renewal activity of HSCs[28], these data suggested that Pot1a might be preventing the production of ROS in HSCs in culture. To test this further, we assessed the relationship between Pot1a expression and ROS production in LT-HSCs (Fig. 5c) and found that overexpression of Pot1a substantially reduced overall levels of intracellular ROS, as well as mitochondrial ROS by comparison with controls (Fig. 5d–f). As activation of the mTOR pathway leads to ROS production[29, 30] we also examined the expression of mTOR (*Mtor*) and Raptor (*Rptor*) mRNA in Pot1a overexpressing or knockdown LT-HSCs (Fig. 5g–i). We found that overexpression of Pot1a substantially decreased *Mtor* and *Rptor* expression (Fig. 5h), while knockdown of Pot1a substantially increased *Mtor* and *Rptor* expression (Fig. 5i). Although the mechanistic details are not currently clear, these data suggest that Pot1a plays a part in preventing the activation of mTOR signaling and production of ROS that may contribute to the maintenance of LH-HSCs.

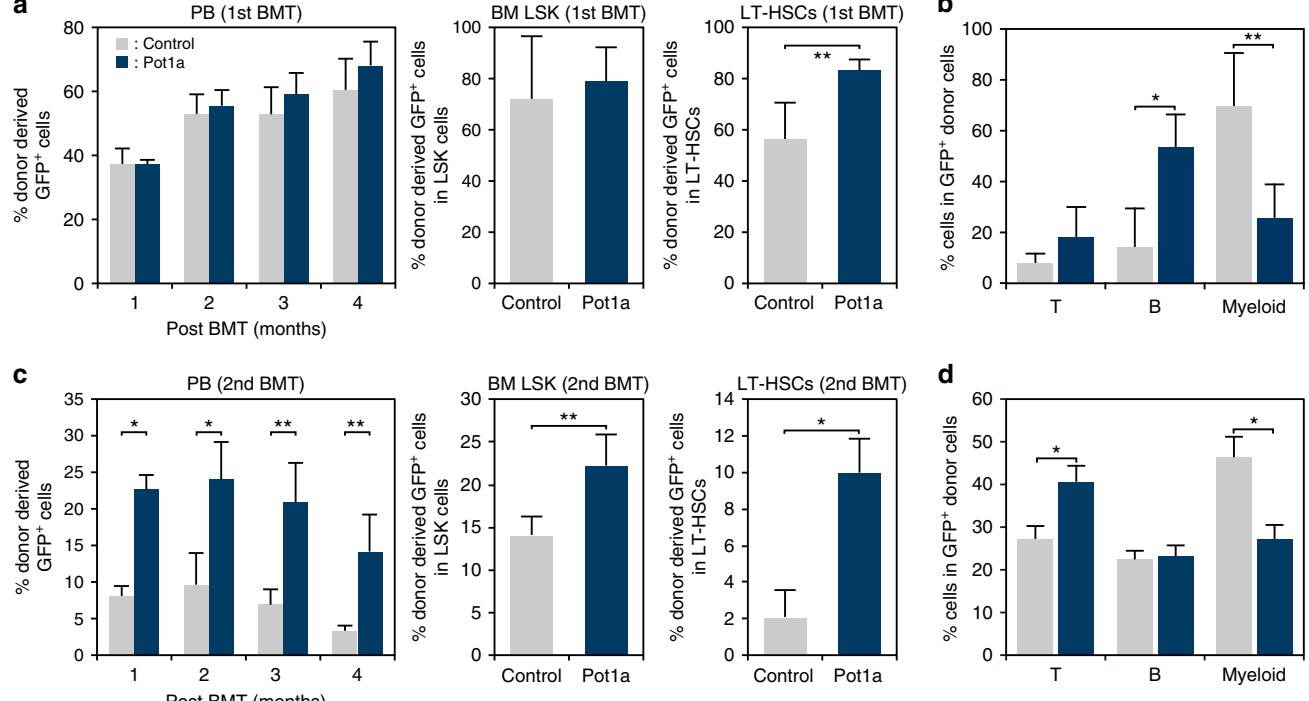

**Fig. 3** Overexpression of Pot1a maintains HSC LTR activity. **a**, **b** Results of primary BMT. **a** Percentage of donor-derived (Ly5.1[+]GFP[+]) cells in recipient mice PB, BM LSK and LT-HSCs 4 months after BMT. **b** Peripheral blood analysis of B, T, and myeloid cell lineages 4 months post BMT. Data are expressed as the mean ± SD ($n = 5$ group[-1], **$p < 0.05$ by $t$-test). Representative data from two independent experiments are shown. **c**, **d** Results of secondary BMT. **c** The percentage of donor-derived (Ly5.1[+]GFP[+]) cells in recipient mice PB, BM LSK and LT-HSCs 4 months post secondary BMT. **d** Peripheral blood analysis of B, T, and myeloid cell lineages 4 months post secondary BMT. Data are expressed as the mean ± SD ($n = 5$, *$p < 0.01$ by $t$-test). Representative data from two independent experiments are shown

**POT1a protein inhibits DDR and maintains HSC in culture.** As Pot1a overexpression is able to maintain HSC activity, we next sought to determine if treatment with exogenous POT1a protein has a similar effect. For this purpose, we prepared a recombinant mouse POT1a protein (MTM-POT1a) with 6xHis and membrane-translocating motif (MTM)-tags (Supplementary Fig. 7a) to facilitate cellular uptake[31, 32]. This method allows Pot1a protein to be delivered without retroviral transduction and therefore does not risk insertional mutagenesis. HSCs incorporated MTM-POT1a at high efficiency (>90%) within 2 h, and the incorporated protein was detectable in LSK cells at least 96 h after culture (Supplementary Fig. 7b, c). Co-staining of Pot1a with TRF1 confirmed that MTM-POT1a was appropriately localized to telomere regions (Supplementary Fig. 7d). Culture of LT-HSCs with MTM-POT1a enhanced CFU-C and HPP-CFC numbers (Fig. 6a). Furthermore, MTM-POT1a increased the numbers of LT-HSCs in culture (Fig. 6b). Interestingly, treatment with MTM-POT1a did not affect cellular survival rate in culture (Fig. 6c), but was able to inhibit Pot1a knockdown-induced apoptosis of LT-HSCs (Fig. 6d). These data suggest that Pot1a prevents differentiation of LT-HSCs in culture. Similarly, in cases when Pot1a expression has been lost (e.g., due to ageing) exogenous Pot1a improves HSC survival. As was the case with Pot1a overexpression, treatment with MTM-POT1a inhibited DDR in LT-HSCs after 3 weeks of culture (Fig. 6e) and treatment with MTM-POT1a substantially increased LT-HSC numbers in culture over the same time period (Fig. 6f). However, cell cycle analysis, using LT-HSCs derived from the Fucci mouse[33], showed that MTM-POT1a did not affect the frequencies of S/G2/M cells in HSPCs in culture (Supplementary Fig. 7e, f). Taken together these results indicate that Pot1a maintains LT-HSC fraction in culture by preventing DNA damage rather than affecting the rate of cell division directly.

**Recombinant POT1a maintains self-renewal activity of LT-HSCs.** We next examined the effect of exogenous POT1a protein on HSC function and sought to assess the extent to which treatment with MTM-POT1a was able to preserve stem cell function ex vivo by performing serial BMTs of control MTM protein and MTM-POT1a treated LT-HSCs. We observed that culture with MTM-POT1a for 10 days prior to transplantation substantially increased the chimerism of donor-derived cells after the first transplantation by comparison with controls (Fig. 7a). PB analysis showed that treatment with MTM-POT1a did not affect the ability of donor-derived cells to differentiate into myeloid, B-cells, and T-cell lineages (Fig. 7b), indicating maintenance of robust multi-lineage reconstitution. In addition, BM cells isolated from recipient mice transplanted with MTM-POT1a treated cells again showed substantially increased chimerism and robust multi-lineage reconstitution (Fig. 7c, d). Importantly, mutant forms of the MTM-POT1a proteins that lack the OB-fold domains (MTM-POT1aΔOB) or TPP1-binding domain (TBD) (MTM-POT1aΔTBD) (Supplementary Fig. 8a, b) did not prevent telomeric DDR in LT-HSCs in culture (Supplementary Fig. 8c, d). In addition, these deletion mutant POT1a proteins did not affect long-term engraftment of HSCs (Fig. 7e), HSC colony forming ability in culture (Supplementary Fig. 8e), telomerase activity in culture (Supplementary Fig. 8f), or LT-HSC numbers in vitro (Supplementary Fig. 8g). To clarify the long-term effect of the overexpression of mutant forms of Pot1a, we transplanted Pot1aΔOB and Pot1aΔTBD transduced LT-HSCs. In contrast to treatment with MTM protein, overexpression of mutant forms of Pot1a inhibited the long-term engraftment of donor HSCs (Fig. 7f). These results indicate that the binding of Pot1a to telomeric DNA as part of the shelterin complex is critical for the maintenance of HSC function.

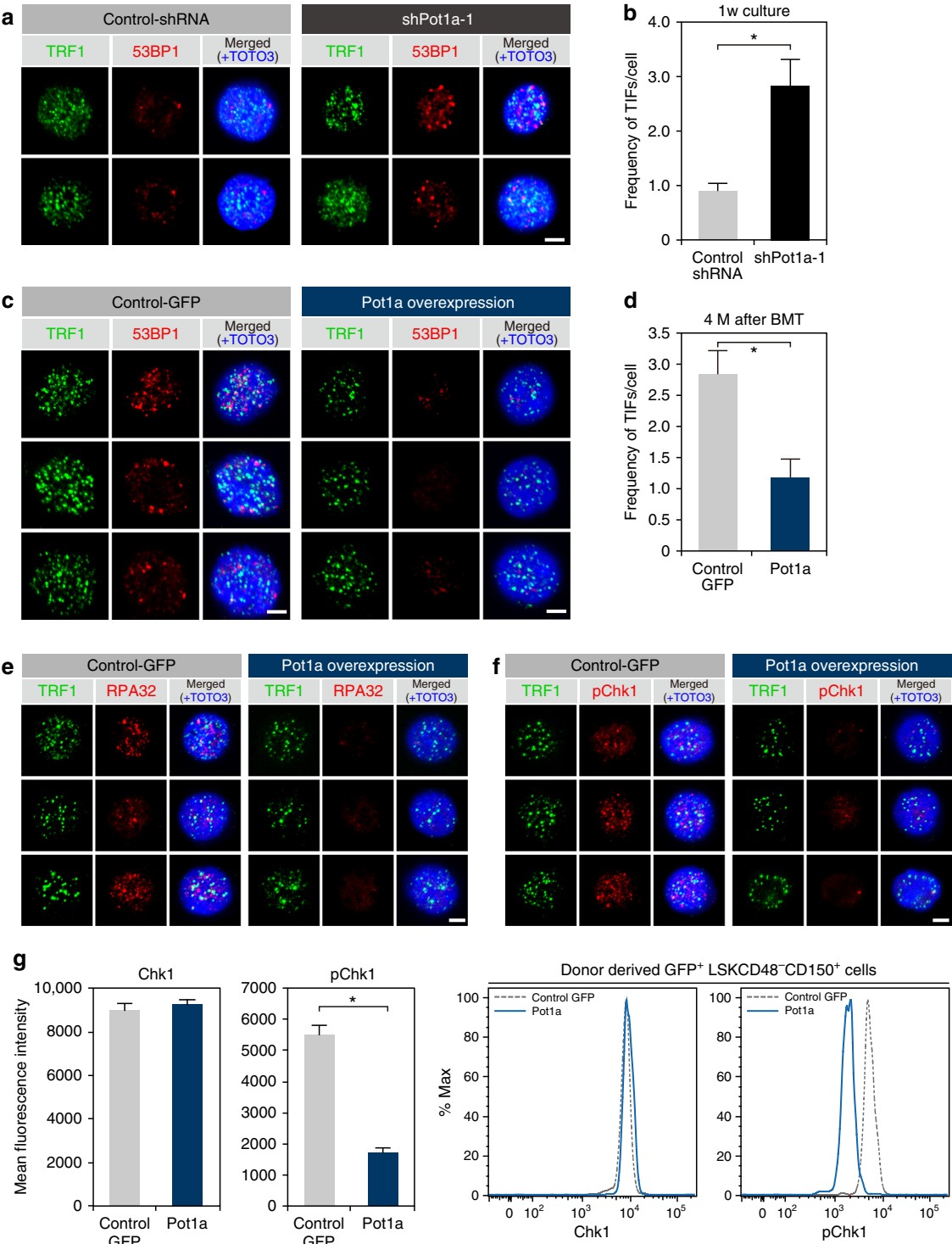

**Fig. 4** Pot1a prevents DDR in HSCs. **a, b** Telomeric DDR in 8 week-old LSK cells upon Pot1a knockdown. **a** Immunocytochemical staining of TRF1 (*green*) and 53BP1 (*red*). Foci co-stained with TRF1 and 53BP1 were identified as TIFs. Nuclei were stained with TOTO3 (*blue*). *Scale bar*, 2 μm. **b** Frequency of TIFs per cell after 1 week of culture. Data are expressed as the mean ± SD ($n = 100$, *$p < 0.01$ by *t*-test). Representative data from three independent experiments are shown. **c–f** Telomeric DDR in donor-derived control-GFP or Pot1a overexpressing 8 week-old LSK cells 4 months post BMT. **c** Immunocytochemical staining of TRF1 (*green*) and 53BP1 (*red*). Nuclei were stained with TOTO3 (*blue*). *Scale bar*, 2 μm. **d** Frequency of TIFs per cell. Data are expressed as the mean ± SD ($n = 120$, *$p < 0.01$ by *t*-test). Representative data from 3 independent experiments are shown. **e** Immunocytochemical staining of TRF1 (*green*) and RPA32 (*red*). **f** Immunocytochemical staining of TRF1 (*green*) and pChk1 (*red*). *Scale bar*, 2 μm. **g** Flow cytometric analysis of Chk1 and pChk1 in donor-derived GFP$^+$ LSKCD48$^-$CD150$^+$ cells after 5 months of 2nd BMT. Mean fluorescence intensity of Chk1 and pChk1 (*left panels*). Data are expressed as the mean ± SD ($n = 3$, *$p < 0.01$ by *t*-test). Representative FACS profiles of Chk1 and pChk1 in donor-derived GFP$^+$ LSKCD48$^-$CD150$^+$ cells (*right panels*)

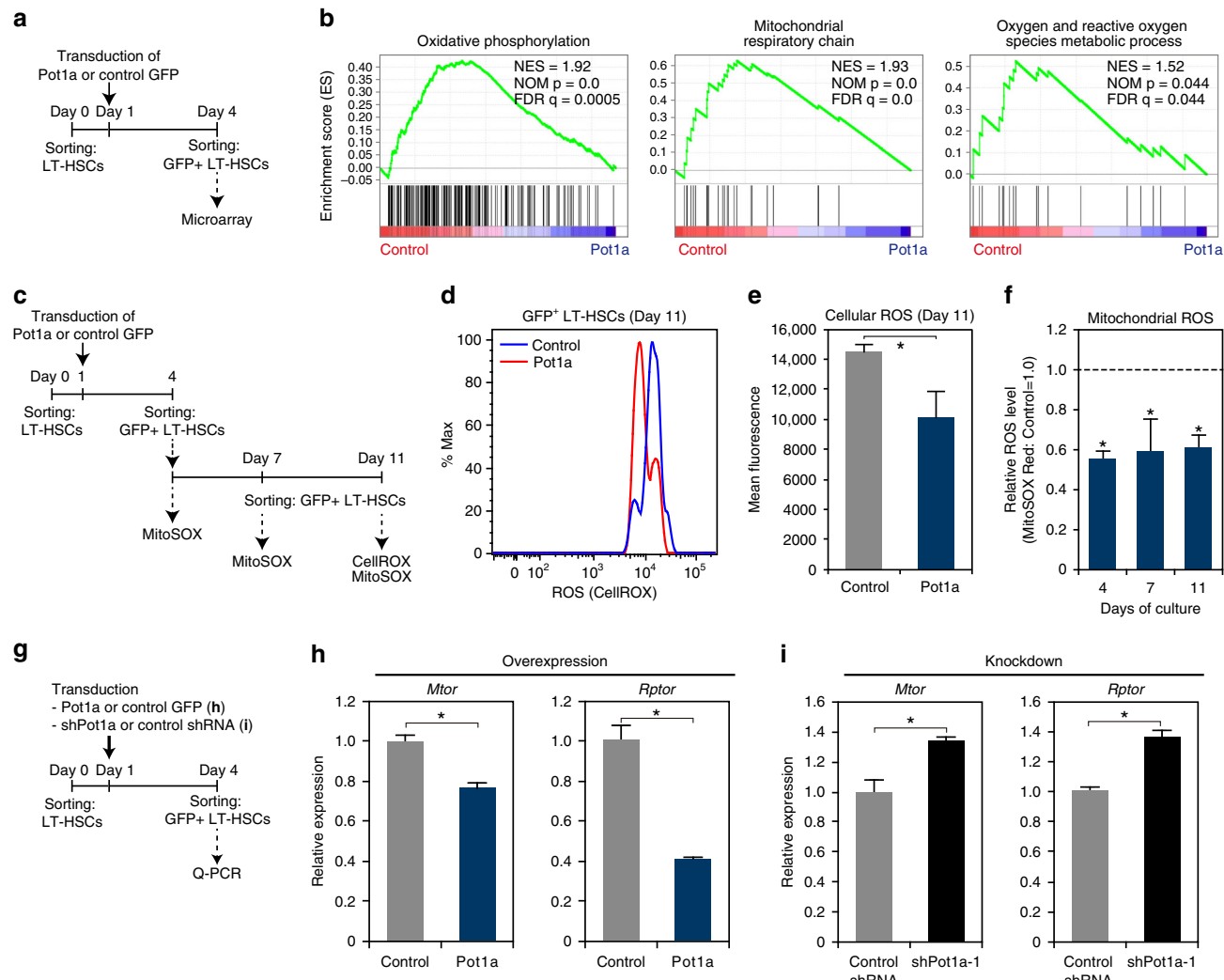

**Fig. 5** Effect of exogenous Pot1a on ROS production in LT-HSCs in culture. **a** Schematic of the microarray analysis. **b** GSEA plots demonstrating enrichment levels of indicated gene sets in control control-GFP versus Pot1a-transduced LT-HSCs. NES, NOM *p* value, and FDR are indicated. **c** Schematic of the measurement of ROS in LT-HSCs. **d** Intracellular ROS levels in control-GFP (*blue*) and Pot1a-transduced (*red*) LT-HSCs. Representative FACS profiles of CellROX in GFP+ LT-HSCs are shown. **e** Mean fluorescence level. Data are expressed as the mean ± SD (*n* = 5, *\*p* < 0.01 by *t*-test). Representative data from two independent experiments are shown. **f** Relative mean fluorescence level of MitoSOX in the GFP+ LT-HSCs on days 4, 7, and 11 of culture. Mean fluorescence level of MitoSOX in control-GFP-transduced LT-HSCs was set to 1.0. Data are expressed as the mean ± SD (*n* = 6: day 7, *n* = 12: day 7, *n* = 3: day 11, *\*p* < 0.01 by Tukey's test). **g** Schematic of the measurement of ROS in LT-HSCs upon overexpression or knockdown of Pot1a. **h** Expression of *Mtor* and *Rptor* in Pot1a-overexpressing LT-HSCs. Data are expressed as the mean ± SD (*n* = 4, *\*p* < 0.01 by *t*-test). **i** Expression of *Mtor* and *Rptor* in Pot1a knockdown LT-HSCs. Data are expressed as the mean ± SD (*n* = 4, *\*p* < 0.01 by *t*-test)

To further verify these results we also conducted limiting dilution competitive BMT. We observed that populations treated with MTM-POT1a had a three-fold enrichment for functional LT-HSCs in comparison with populations treated with control MTM protein (Fig. 7g). Taken together these data suggest that treatment with exogenous Pot1a enhances the self-renewal activity of HSCs in culture.

**Exogenous POT1a improves the activity of aged LT-HSCs.** To investigate the activity of Pot1a during aging we next analyzed the effect of MTM-POT1a on the prevention of TIF formation in aged LT-HSCs. As expected we found that aged (> 90 week-old) LT-HSC had more TIFs than young (8 week-old) LT-HSCs, yet importantly treatment with MTM-POT1a reduced the number of TIFs in cells isolated from both young and old mice after 10 days of culture (Supplementary Fig. 9). We also examined the effect of

MTM-POT1a treatment on the proliferation and apoptosis of aged LT-HSCs. In accordance with the effect that we observed in young LT-HSCs (Fig. 6b, c), treatment of MTM-POT1a increased aged LT-HSC numbers in culture without affecting the total cell number and apoptosis compared with control MTM protein treatment (Fig. 8a–c).

To determine the potency of these effects we sought to assess if treatment with Pot1a was able to restore the activity of aged HSCs in culture. To do this, we compared the reconstitution activity following BMT of LT-HSCs isolated from young (8 week-old) and aged (> 90 week-old) after culture with MTM-POT1a or control MTM protein. As expected, after 10 days of culture the reconstitution activity of aged HSCs was substantially lower than that of young HSCs (Fig. 8d). However, MTM-POT1a treatment raised the engraftment ability of aged HSCs compared with control MTM treatment (Fig. 8d). In addition, aged controls showed a decrease in B-cell and an increase in myeloid cell

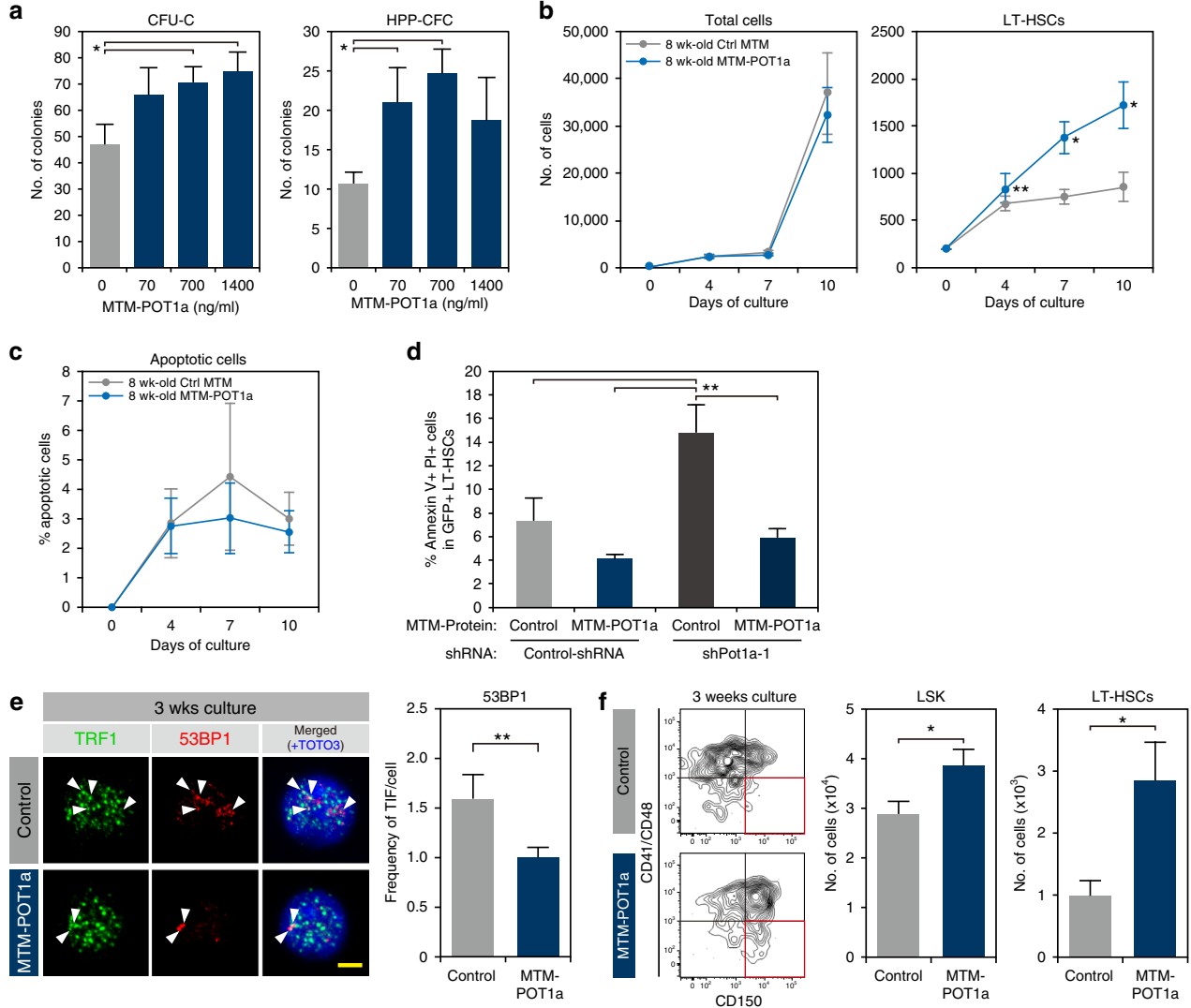

**Fig. 6** Treatment with exogenous POT1 protein protects LT-HSCs in culture. **a** Effect of MTM-POT1a on colony formation of HSCs. LT-HSCs (8 week-old) were cultured with MTM-POT1a for 2 weeks. After culture, Lin⁻ cells were isolated and re-cultured in MethoCult™ GF M3434 medium (200 cells per dish). Data are expressed as the mean ± SD ($n = 3$, *$p < 0.01$ by Tukey's test). Representative data from two independent experiments are shown. **b**, **c** LT-HSCs (8 week-old) were cultured with MTM-POT1a or control MTM protein. **b** Number of total cells and LT-HSCs on day 4, 7, and 10 of culture. Data are expressed as the mean ± SD ($n = 6$, *$p < 0.01$ by $t$-test). **c** Percentage of Annexin V⁺PI⁺ apoptotic cells in LT-HSCs on day 4, 7, and 10 of culture. Data are expressed as the mean ± SD ($n = 6$, *$p < 0.01$ by $t$-test). **d** LT-HSCs (8 week-old) were transduced with control shRNA or shPot1a-1. After 2 days of shRNA transduction, GFP⁺LT-HSCs were re-sorted and cultured with MTM-POT1a or control MTM protein. After 1 week of culture, Annexin V assay was performed. Percentage of Annexin V⁺PI⁺ apoptotic cells in GFP⁺LT-HSC fraction is shown. ($n = 3$, **$p < 0.05$ by Tukey's test). **e** Immunocytochemical staining of TRF (*green*), 53BP1 (*red*), and TOTO3 (*blue*) in 8 week-old LT-HSCs cultured for 3 weeks (*left*). Frequencies of TIFs after 3 weeks of culture (*right*). Data are expressed as the mean ± SD ($n = 80$–100, *$p < 0.01$ by $t$-test). **f** Representative FACS profiles of 8 week-old LT-HSCs after 3 weeks of culture with control MTM protein or MTM-POT1a (*left*). Numbers of LSK cells and LT-HSCs after 3 week culture (starting from 2400 cells) are shown in right panels. Data are expressed as the mean ± SD ($n = 3$, *$p < 0.01$ by $t$-test). Representative data from five independent experiments are shown

differentiation, while treatment with MTM-POT1a diminished this bias (Fig. 8e).

To further understand the function of exogenous Pot1a in the activation of aged HSCs, we performed microarray analysis of control and MTM-POT1a treated aged HSCs. GSEA revealed that aged HSCs cultured with control MTM protein had a tendency to express myeloid-associated genes in culture, while MTM-POT1a treated aged HSCs did not (Fig. 8f). Conversely, MTM-POT1a treated HSCs significantly overexpressed genes associated with long-term HSC identity (Fig. 8f). These results suggest that MTM-POT1a treatment inhibits myeloid differentiation and helps maintain the undifferentiated LT-HSC state.

In addition, similar to our observations in 8 week-old LT-HSCs, we found that genes associated with oxidative phosphorylation, the mitochondrial respiratory chain, Myc targets, E2F targets, DNA repair, and DNA replication were suppressed in aged HSCs cultured with MTM-POT1a by comparison with control MTM treated aged HSCs (Supplementary Fig. 10a). Furthermore, as with young LT-HSCs, overexpression of Pot1a significantly decreased *Mtor* and *Rptor* expression (Supplementary Fig. 10b) and reduced the production of ROS in aged LT-HSCs during culture (Supplementary Fig. 10c-e).

**POT1 expands human LT-HSC numbers**. On the basis of the functional effect of MTM-POT1a on mouse HSCs, we sought to

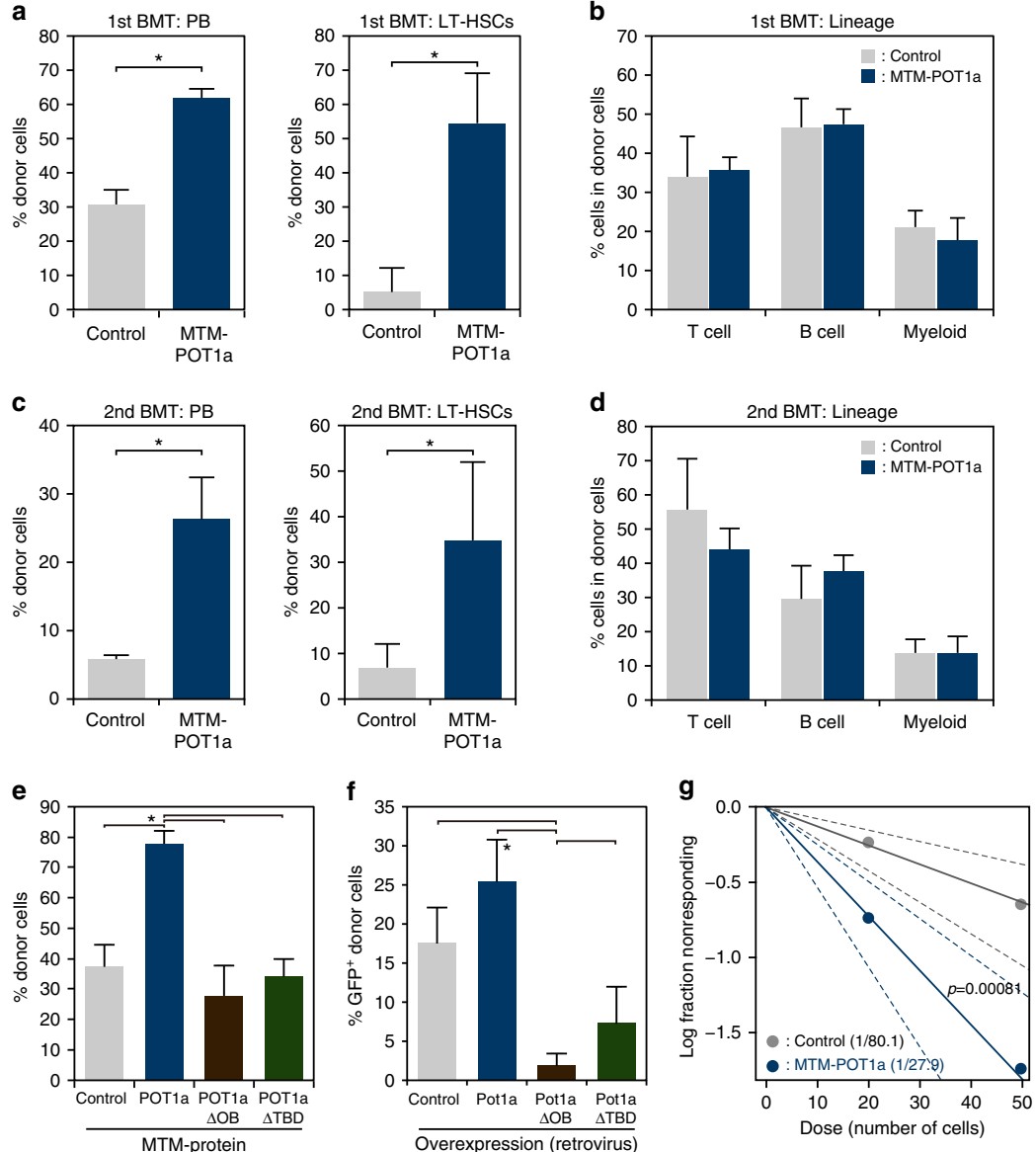

**Fig. 7** MTM-POT1a maintains self-renewal activity of HSCs. **a** Results of primary BMT. Percentage of donor-derived (Ly5.1[+]) cells in PB and LT-HSC fractions 4 months after BMT. **b** Frequency of donor cells in T, B, and myeloid cell fractions in PB 4 months post BMT. Data are expressed as the mean ± SD ($n = 5$ group[−1], *$p < 0.01$, **$p < 0.05$ by $t$-test). Representative data from four independent experiments are shown. **c** Results of secondary BMT. Percentage of donor-derived (Ly5.1[+]) cells in PB and LT-HSC fractions 4 months post secondary BMT. **d** Frequency of donor cells in T, B, and myeloid cell fractions in PB 4 months post secondary BMT. Data are expressed as the mean ± SD ($n = 5$ per group, *$p < 0.01$ by $t$-test). Representative data from four independent experiments are shown. **e** Effect of mutant forms of POT1a in the regulation of LTR activity of HSCs. Percentage of donor-derived (Ly5.1[+]) cells in PB 4 months after BMT is shown. Data are expressed as the mean ± SD ($n = 5$ group[−1], *$p < 0.01$ by Tukey's test). Representative data from two independent experiments are shown. **f** Effect of the overexpression of mutant forms of Pot1a on the maintenance of LTR activity of HSCs. Percentages of GFP[+] donor-derived cells in PB 5 months after BMT are shown. Data are expressed as the mean ± SD ($n = 5$ per group, *$p < 0.01$ by Tukey's test). Representative data from two independent experiments are shown. **g** Result of limiting dilution competitive BMT. A log-fraction plot of the limiting dilution model is shown. The slope of the line is the log-active cell fraction. The dotted lines give 95% confidence intervals ($n = 24$: control 20 cells, $n = 21$: control 50 cells, $n = 23$: MTM-POT1a 20 cells, $n = 23$: MTM-POT1a 50 cells). Competitive repopulation units are also shown

determine whether exogenous human POT1 similarly affects human HSC activity. We observed that human cord blood (hCB)-derived Lin[−]CD34[+]CD38[−]CD90[+]CD45RA[−]CD49f[+] LT-HSCs[34] expressed significantly higher levels of *POT1* than CD34[+] HSPCs (Fig. 9a). To assess the effect of POT1 on human HSC activity we prepared MTM-POT1 recombinant protein (Supplementary Fig. 7a) and cultured human CB and BM LT-HSCs with MTM-POT1 for 10 days. In accordance with the role of MTM-POT1a in mouse, MTM-POT1 significantly increased the number of CFU-Cs and HPP-CFCs from hCB LT-HSCs (Fig. 9b) and

markedly reduced total and telomeric DDR (Fig. 9c, d). Although it did not affect the total cell number or number of CD34[+]CD38[−] cells, treatment with MTM-POT1 significantly increased the number of LT-HSCs in culture of CB-derived HSCs (Fig. 10a, b) but did not influence LT-HSC apoptosis (Fig. 10c). Similarly, the treatment of BM-derived LT-HSCs with MTM-POT1 also increased the number of Lin[−]CD34[+]CD38[−] cells and Lin[−]CD34[+]CD38[−]CD90[+]CD45RA[−]CD49f[+] LT-HSCs in cultures (Fig. 10d). In accordance with these results we observed that populations treated with MTM-POT1 contained 4.5-fold more functional

LT-HSCs in comparison with controls following limiting dilution BMT (Fig. 10e). Collectively, these data indicate that POT1 regulates human HSC activity in a similar manner to that of Pot1a in mice, and may be used to efficiently maintain human HSC activity ex vivo.

## Discussion

Taken together our data demonstrate that Pot1a has a critical role in maintaining HSC activity during long-term in vitro culture or subsequent to BMT. We found that Pot1a regulates HSC activity by inhibiting ATR-dependent telomeric DNA damage, and thereby protecting cells from associated apoptosis. These results indicate that the formation of the shelterin complex at the telomeric region is important to Pot1a mediated maintenance of LT-HSC activity. However, in addition to this telomeric role we have also identified a novel non-telomeric role. This new non-telomeric role is particularly interesting since reduction of ROS is thought to be crucial in inhibiting global DNA damage in

LT-HSCs in culture. Energy metabolism in quiescent HSCs largely depends on glycolysis, while activated HSCs synthesize adenosine-5′-triphosphate (ATP) mainly through oxidative phosphorylation[35, 36]. Our finding that HSCs upregulate genes associated oxidative phosphorylation in culture suggests that culture induces a metabolic shift from glycolysis to oxidative phosphorylation that is characteristic of stress-induced activation, yet this shift can be inhibited by Pot1a treatment. Moreover, we observed that Pot1a expression is negatively associated with mTOR expression in HSCs. This suggests that Pot1 may have an important role in regulating mTOR signaling, although further work to dissect the details of this non-telomeric role for Pot1a in the maintenance of HSC activity is needed.

In addition to its role in protecting against stress we also found that Pot1a has a central role in regulating stem cell activity during aging. We observed that expression of Pot1a is lost during aging, and this loss results in the accumulation of DNA damage, alterations in metabolism and an increase in ROS production, which in turn compromises aged HSC function. However,

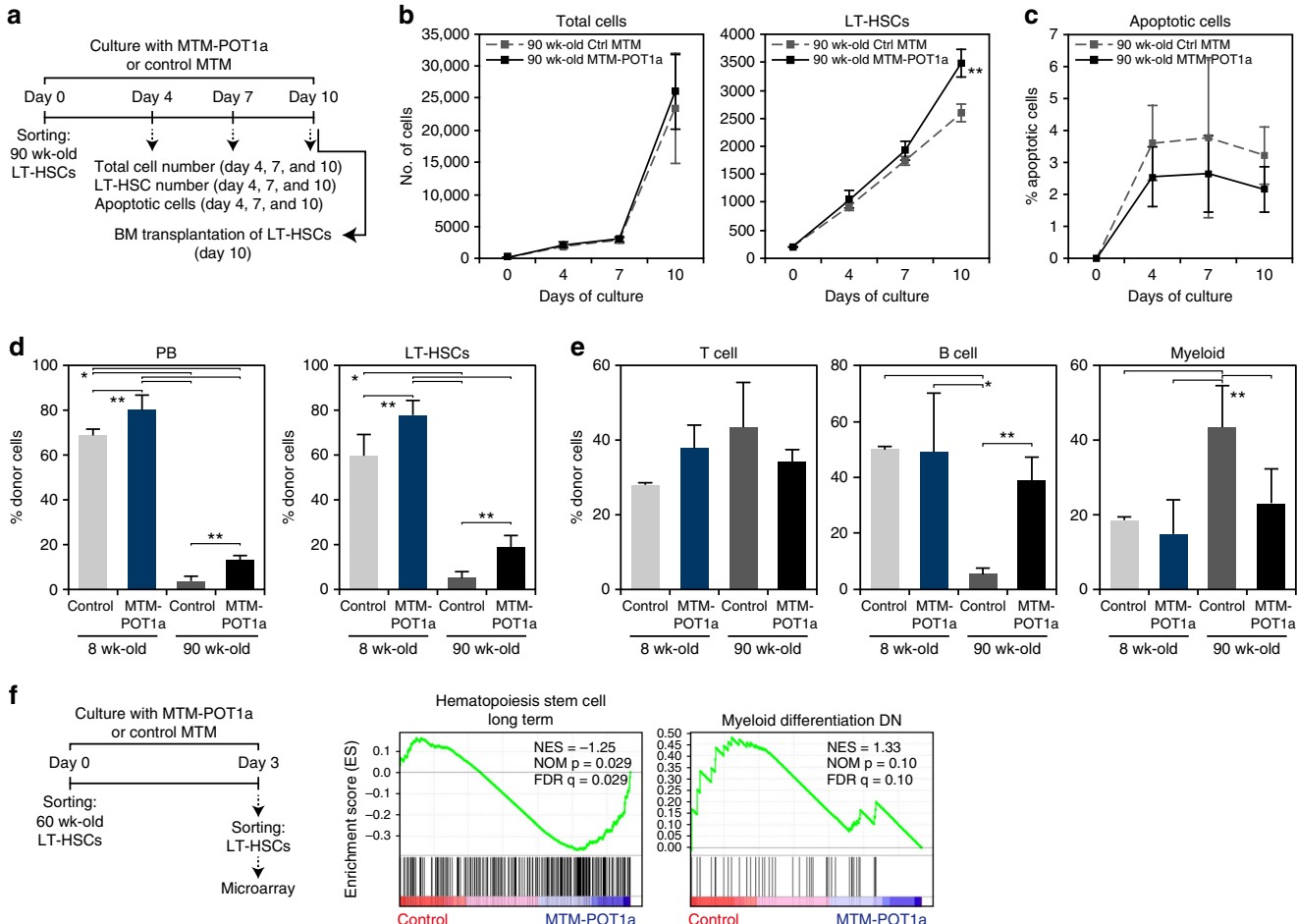

**Fig. 8** MTM-POT1a improves aged HSC function. **a** Schematic of the analysis of cell number and apoptotic fraction of 90 week-old LT-HSCs after culture with MTM-POT1a or control MTM protein. **b** Number of total cells and LT-HSCs on days 4, 7, and 10 of culture. Data are expressed as the mean ± SD ($n = 5$: day 4, $n = 6$: day 7 and 10, **$p < 0.05$ by $t$-test). **c** Percentage of Annexin V[+]PI[+] apoptotic cells in LT-HSC fraction on day 4, 7, and 10 of culture. Data are expressed as the mean ± SD ($n = 6–8$). **d, e** Effect of MTM-POT1a on reconstitution after BMT of young and aged LT-HSCs. 8 and 90 week-old LT-HSCs were cultured with MTM-POT1a or control MTM protein for 10 days and transplanted into lethally irradiated mice. **d** Percentages of donor-derived (Ly5.1[+]) cells in PB and LT-HSCs 4 months after BMT are shown ($n = 5$ per group, *$p < 0.01$, **$p < 0.05$ by Tukey's test). Representative data from two independent experiments are shown. **e** Percentage of T cells (CD3[+]), B cells (B220[+]), and myeloid (Mac-1[+]/Gr-1[+]) cells in donor-derived cells in PB ($n = 5$ group[−1], *$p < 0.01$, **$p < 0.05$ by Tukey's test). Representative data from 2 independent experiments are shown. **f** 60 week-old LT-HSCs were cultured with MTM-POT1a or control MTM protein for 3 days. After the culture, gene expression profiles were analyzed by microarray. GSEA plots demonstrating enrichment levels of indicated gene sets in control MTM protein treated cells versus MTM-POT1a treated cells. NES, NOM $P$ value, and FDR are indicated

we observed that this decline is reversible: remarkably ex vivo treatment of aged LT-HSCs with recombinant POT1a is able to re-activate aged HSCs. Since Pot1a overexpression inhibited the expression of *Mtor* and *Rptor* in aged LT-HSCs, the regulation of mTOR signaling by Pot1a may participate in this re-activation of aged HSC function. Although the precise mechanisms by which this functional improvement occurs have yet to be fully determined, our results indicate that exogenous Pot1a can both prevent telomeric and non-telomeric DNA damage and inhibit ROS production, thereby inducing a more potent immature phenotype in aged HSCs upon ex vivo culture. It will be interesting to clarify how these mechanisms are related to one another and determine, for example, whether telomere insufficiencies precede metabolic changes and ROS production or vice versa.

As a similar protective role for POT1 was also seen in human HSCs these results have important implications for the culture of human HSCs ex vivo. Expansion of HSC numbers requires the stimulation of self-renewal divisions[37]. Although a number of strategies, including culture with recombinant proteins[38–41], modulation of intrinsic factors, such as Hoxb4[42–44], Prostaglandin E2[45], or aryl hydrocarbon receptor antagonist[46, 47] have been reported to contribute to HSC expansion in culture, these treatments primarily work by activating the cell cycle. By contrast, we find that Pot1a does not directly activate the cell cycle of HSCs, but rather protects HSCs from DNA damage under stress. On the basis of these findings, we anticipate that, in combination with current methods, POT1 will be of significant use in the development of robust and safe methods for ex vivo expansion of human HSCs.

Taken together our results highlight general telomeric and non-telomeric mechanisms by which Pot1 regulates stem cell activity in vitro and in vivo, with widespread implications for our understanding of age-related degeneration and applications in regenerative medicine.

## Methods

**Mice and cells**. C57BL/6 (B6-Ly5.2), C57BL/6 mice congenic for the Ly5 locus (B6-Ly5.1) were purchased from Sankyo-Lab Service (Tsukuba, Japan). Fucci mice were provided by Dr. Miyawaki (Brain Science Institute, RIKEN, Wako-city, Saitama, Japan). Tert mutant mice were provided by Dr. Ishikawa (Kyoto University, Kyoto, Japan). Female NOD/SCID/IL-2Rc[null] (NOG) mice were purchased from the Central Institute for Experimental Animals (Chiba, Japan) and bred in a pathogen-free environment. In this study, female mice were used for all experiments except the microarray analysis. hCB-derived CD34[+] cells (catalog number: 2C-101B) and hBM-derived CD34[+] cells (catalog number: 2M-101) were purchased from Lonza. The Gene recombination experiment safety committee and animal experiment committee in both Keio University and Kyushu University approved this study and all experiments were carried out in accordance with the Guidelines for Animal and Recombinant DNA experiments at Keio University and Kyushu University.

**Mouse antibodies**. The following monoclonal antibodies (Abs) were used for flow cytometry and cell sorting: anti-c-Kit (2B8, BD Biosciences, 1:100), -Sca-1 (E13-161.7, BD Biosciences, 1:100), -CD3e (145-2C11, BD Biosciences, 1:100), -CD4 (RM4-5, BD Biosciences, 1:100), -CD8 (53-6.7, BD Biosciences, 1:100), -B220 (RA3-6B2, BD Biosciences, 1:100), -TER-119 (BD Biosciences, 1:100), -Gr-1 (RB6-8C5, BD Biosciences, 1:100), -Mac-1 (M1/70, BD Biosciences, 1:100), -Flt3 (A2F10.1, BD Biosciences, 1:100), -CD41 (MWReg30, eBioscience, 1:100), -CD48 (HM48-1, Biolegend, 1:100), -CD150 (TC15-12F12.2, Biolegend, 1:100), -CD45.1 (A20, BD Biosciences, 1:100), -CD45.2 (104, BD Biosciences, 1:100), -CD45 (30-F11, BD Biosciences, 1:100), and -Ki67 (B56, BD Biosciences, 1:30). A mixture of CD4, CD8, B220, TER-119, Mac-1, and Gr-1 was used as the lineage mix. Anti-Chk1 (sc-8408, Santa Cruz Biotechnology, 1:100), anti-Phospho Chk1 (133D3, Cell Signaling Technology, 1:100), and anti-53BP1 (NB100-304,

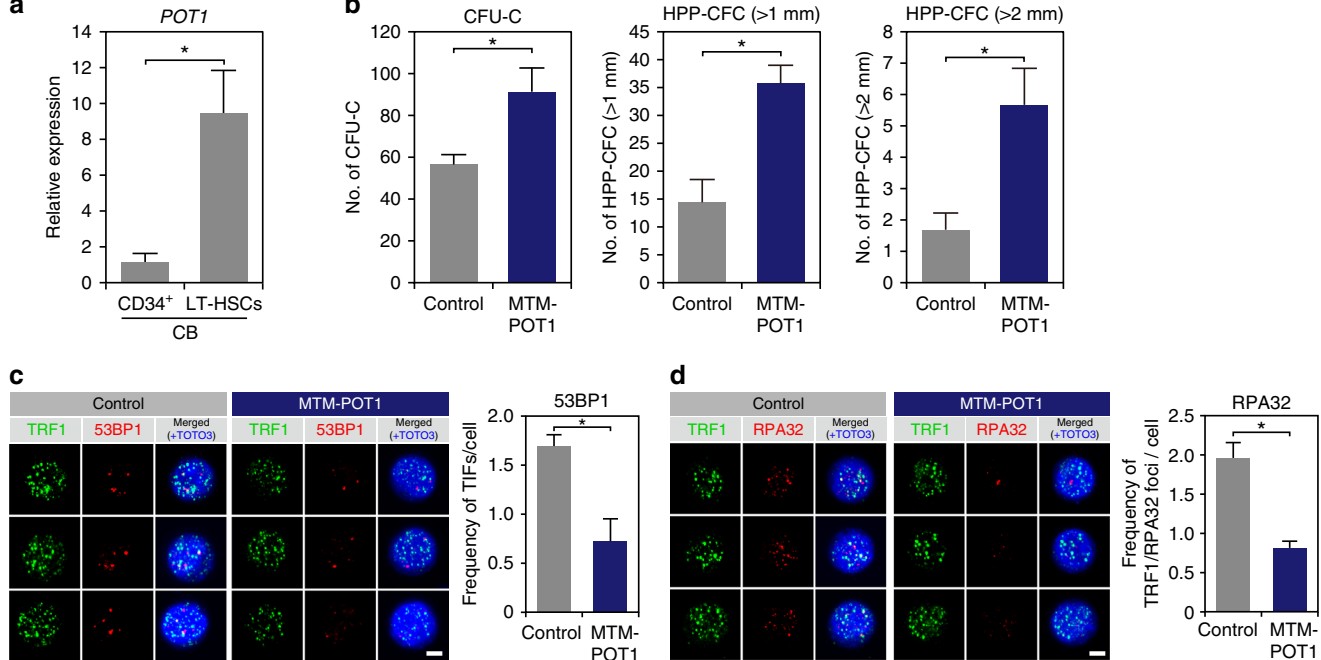

**Fig. 9** POT1 prevents DDR in human HSC. **a** Expression of POT1 in hCB CD34[+] cells and LT-HSCs (Lin[−]CD34[+]CD38[−]CD45RA[−]CD90[+]CD49f[+]). **b** hCB-derived LT-HSCs were cultured with MTM-POT1. After 10 days of culture, cells were isolated and re-cultured in methylcellulose medium (200 cells per dish). The number of CFU-C and HPP-CFC (>1.0 mm, >2.0 mm) are shown. Data are expressed as the mean ± SD (n = 3, *p < 0.01 by t-test). Representative data from three independent experiments are shown. **c**, **d** hCB LT-HSCs were cultured for 10 days with control MTM protein or MTM-POT1. After 10 days of culture, LT-HSCs were re-isolated and number of TIF was examined. **c** Immunocytochemical staining of TRF1 (*green*), 53BP1 (*red*), and TOTO3 (*blue*). Scale bar, 2 μm (*left*). Frequencies of TIFs after 10 days of culture (*right*). Data are expressed as the mean ± SD (n = 100: control, n = 100: MTM-POT1, *p < 0.01 by t-test). Representative data from 2 independent experiments are shown. **d** Immunocytochemical staining of TRF1 (*green*), RPA32 (*red*), and TOTO3 (*blue*). Scale bar, 2 μm (*left*). Frequencies of TIFs after 10 days of culture (*right*). Data are expressed as the mean ± SD (n = 110–120: control, n = 110: MTM-POT1, *p < 0.01 by t-test). Representative data from two independent experiments are shown

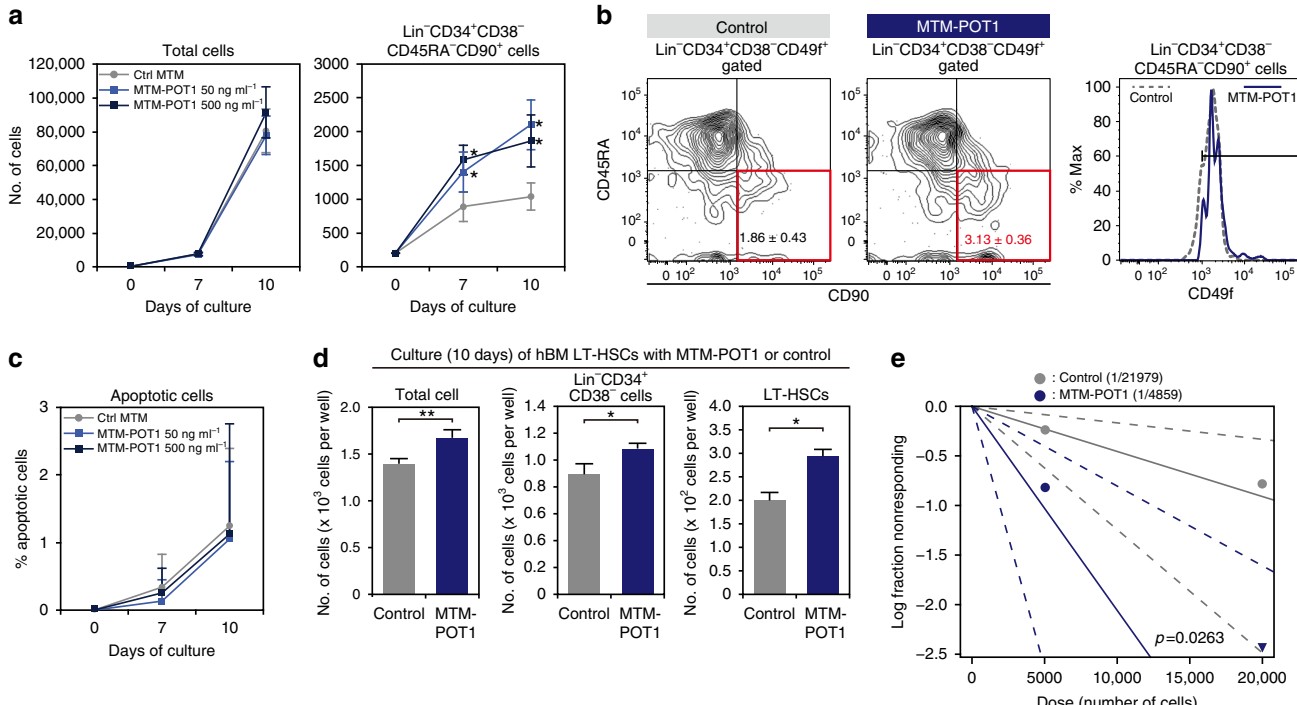

**Fig. 10** POT1 maintains self-renewal activity of human HSCs. hCB-derived LT-HSCs (200 cells per well) were cultured for 10 days with or without MTM-POT1. **a** Number of total cells (*left*) and Lin⁻CD34⁺CD38⁻CD45RA⁻CD90⁺ cells on day 7 and 10 of culture. Data are expressed as the mean ± SD ($n = 10$, *$p < 0.01$ by *t*-test). **b** Representative FACS profiles of CD90 and CD45RA in Lin⁻CD34⁺CD38⁻CD49f⁺ cells (*left*) and the expression of CD49f in the Lin⁻CD34⁺CD38⁻CD45RA⁻CD90⁺ fraction (*right*). **c** Percentage of Annexin V⁺PI⁺ apoptotic cells in Lin⁻CD34⁺CD38⁻CD45RA⁻CD90⁺ cells on day 7 and 10 of culture. Data are expressed as the mean ± SD ($n = 11$: day 7, $n = 9$: day 10). **d** LT-HSCs isolated from hBM (200 cells per well) were cultured for 10 days with or without MTM-POT1 (50 ng ml⁻¹). Number of total cells, CD34⁺CD38⁻ cells, and LT-HSCs after culture are shown. Data are expressed as the mean ± SD ($n = 3$, *$p < 0.01$, **$p < 0.05$ by *t*-test). Representative data from two independent experiments are shown. **e** Result of limiting dilution BMT. A log-fraction plot of the limiting dilution model is shown. The slope of the line is the log-active cell fraction. The dotted lines give 95% confidence intervals ($n = 5$). The data point with zero negative response at dose 20,000 is represented by a downward-pointing triangle. Estimated numbers of repopulating cells are also shown

Novus biologicals, 1:100) were used for the analysis of the phosphorylation of Chk1 and 53BP1 by FACS. The following Abs were used for immunocytochemistry and immunoblotting: anti-POT1 (ab21382, Abcam for immunocytochemistry, 1:200. sc-27951, Santa Cruz Biotechnology for immunoblotting, 1:1000), anti-53BP1 (NB100-304, 1:200), anti-TRF1 (sc-5475, Santa Cruz Biotechnology, 1:200), anti-RPA32 (sc-28709, Santa Cruz Biotechnology, 1:200), anti-phospho Chk1 (ab47318, Abcam, 1:200), and anti-His-Tag (ab27025, Abcam, 1:200).

**Human antibodies**. The following monoclonal Abs were used for flow cytometry and cell sorting: anti-CD34 (581, BD Biosciences, 1:5), -CD38 (HIT2, BD Biosciences, 1:5), -CD45RA (HI100, BD Biosciences, 1:20), -CD90 (5E10, BD Biosciences, 1:20), -CD49f (GoH3, Biolegend, 1:5), and -CD45 (HI30, BD Biosciences, 1:20). A mixture of biotin-conjugated anti-CD2, -CD3, -CD11b, -CD14, -CD15, -CD16, -CD19, -CD56, -CD123, and -CD235a (130-092-211, Lineage Cell Depletion Kit human, Miltenyi Biotec Inc., 1:5) was used as the lineage mix. The following Abs were used for immunocytochemistry: anti-human TRF1 (TRF-78, ab10579, Abcam, 1:100), 53BP1 (NB100-304, 1:200), and RPA32 (sc-28709, 1:200).

**Cell preparation and flow cytometry**. BM cells were isolated from femurs and tibias by flushing with PBS. Mononuclear cells (MNCs) were isolated by centrifugation of total BM cells on Lymphoprep™ (Alere Technologies AS). For isolation of mouse HSCs, c-Kit⁺ cells were enriched by magnetic cell sorting with anti-CD117 beads (130-091-224, Miltenyi Biotec Inc. 1:5 dilution). c-Kit enriched cells were stained with lineage marker Abs (anti-CD4, -CD8, -B220, -Gr1, -Mac-1, -Ter119, anti-Sca-1, anti-c-Kit, anti-CD48, and anti-CD150). For isolation of human HSCs, hCB, and hBM CD34⁺ cells were stained with a mixture of biotin-conjugated lineage Abs (anti-CD34, -CD38, -CD45RA, -CD90, -CD49f) and a fluorochrome labeled streptavidin (1:100 dilution). Stained cells were analyzed and sorted using a FACSAria (BD Biosciences).

**Immunocytochemistry**. Cells were spread onto glass slides and fixed in 4% paraformaldehyde/PBS(−) or methanol/acetone. After blocking, samples were

incubated at 4 °C overnight with a primary Ab. After washing with PBS, samples were stained with florescence-labeled secondary Ab. For nuclear staining, specimens were treated with TOTO3 or DAPI (Molecular Probes, 1:300). Fluorescence images were examined using a confocal laser-scanning microscope (FV1000, Olympus or LSM700, Zeiss).

**Image analysis**. Assessment of co-localization of staining for TRF1 (green) and 53BP1 or RPA32 (red) was conducted, as previously described[48, 49]. Briefly, subsequent to thresholding using Otsu's method[50], the extent of co-localization for each image was determined using Manders' co-localization coefficients

$$M1 = \frac{\sum_i G_{i,\text{colocal}}}{\sum_i G_i}$$

$$M2 = \frac{\sum_i R_{i,\text{colocal}}}{\sum_i R_i}$$

where $R_i$ and $G_i$ are the intensities of the $i$th pixel in the red and green channels respectively, and $R_{i,\text{colocal}} = R_i$ if $G_i > 0$ and zero otherwise (and similarly for $G_{i,\text{colocal}}$). Significance of co-localization was obtained by comparing observed co-localization coefficients with those obtained by independently scrambling the red and green channels $1 \times 10^4$ times. Note that since the number of red and green pixels may differ, M1 and M2 are not generally equal. Random scrambling of pixels does not preserve local spatial correlations in pixel intensities and can lead to overestimation of the significance of colocalization[48]. Therefore, the binary images from both channels were randomly divided into blocks approximately the size of a TIF and these blocks (rather than individual pixels) were scrambled, as previously described[48]. This method preserves local spatial correlations in pixel intensities and therefore provides a conservative null model. Significance of co-localization was determined by calculating the proportion of times that randomization increased the overlap coefficients M1 and M2 relative to the coefficients obtained before scrambling and by comparing observed and randomized co-localization coefficients using Welch's paired *t*-test. All analysis was restricted to regions identified as cell nuclei using co-staining for TOTO3 (blue). In total co-localization in > 3750 cells from > 1000 independent images were assessed.

**Immunoblot analysis**. To evaluate Pot1 expression levels in hematopoietic stem/progenitor cells, BM Lin+, Lin−, LSK150−, and LSKCD150+ cells (1 × 10^5 cells) were lysed and subjected to SDS–PAGE and immunoblotting using antibodies against POT1 polyclonal Ab (Santa Cruz Biotechnology, Inc.) (1:1000). Rabbit anti-Goat IgG HRP (DAKO) (1:5000) was used for the secondary Ab. Densitometric quantification was performed on scanned immunoblot images using Multi Gauge V3.1 (Fuji Film). To evaluate the efficiency of Pot1a knockdown, LT-HSCs were transduced with retrovirus expressing control-sRNA, shPot1a-1 or −2. After 2 days of the transduction of shRNA, GFP+ LSKCD150+ (1 × 10^5 cells) were isolated and analyzed Pot1 expression by western blot. All uncropped western blots can be found in Supplementary Fig. 11.

**Q-PCR analyses**. Q-PCR analysis was performed on an ABI 7500 Fast Real-Time PCR System using TaqMan Fast Universal PCR master mixture (Applied Biosystems, Foster City, CA, USA). The TaqMan® Gene Expression Assay mixes used in this study are listed below. Data were analyzed by 7500 Fast System SDS Software 1.3.1. All experiments were carried out in triplicate.

**Q-PCR array analysis**. Gene expression was analyzed using the BioMark 96·96 Dynamic Array (Fluidigm). Data were analyzed using BioMark Real-Time PCR Analysis Software Version 3.0.2 (Fluidigm). The TaqMan Gene Expression Assay Mixes used in this study are listed below.

**TaqMan® gene expression assay mixes**. TaqMan® Gene Expression Assays used in this study are listed in Supplementary Tables 2 and 3.

**Intracellular flow cytometry**. For the staining of intracellular antigen, cells were first stained with cell surface markers prior to fixation and permeabilization. Cells were then fixed and permeabilized with Cytofix/Cytoperm™ (BD Bioscience) and stained with Abs against the intracellular antigen. Apoptosis was analyzed by Annexin V assay Kit (BioLegend). ROS level was measured using CellROX® Reagents (Thermo Fisher Scientific) and MitoSOX™ Red (Thermo Fisher Scientific).

**Quantification of telomere length and telomere FISH**. Telomere length was quantified by flow cytometry using flow FISH with a Telomere PNA Kit/FITC for flow cytometry (Dako Cytomation), according to the manufacturer's instructions. In brief, Pot1a or control-mKO1 transduced LSK cells were divided into two tubes, denatured in hybridization solution with or without telomere-PNA probes, and hybridized overnight. After washing and DNA staining, the labeled or non-labeled (control) samples were analyzed by FACS.

**Quantitative telomeric repeat amplification protocol**. Telomerase activity was measured with a Q-TRAP assay and used a Quantitative Telomerase Detection Kit (US Biomax, Inc.). In this assay, sorted cells were lysed, and the lysate was added to telomeric repeats onto a substrate oligonucleotide. The resultant extended product was subsequently amplified by PCR. Direct detection of PCR product was monitored on an ABI 7500 Fast Real-Time PCR System.

**Sequences of Pot1a shRNAs**. The sequences of Pot1a shRNAs were as follows: shPot1a-1, 5′-GGAAGGTACAATTGCCAAT-3′; shPot1a-2, 5′-GCCTCCGTATG TTAGCAAA-3′. Sequences were separated by a nine-nucleotide non-complementary spacer (TTCAAGAGA) from the corresponding reverse complement of the same 19-nucleotide sequence. A scrambled sequence (5′-GACACGCGAC TTGTTGTACCAC-3′) served as a control.

**Construction of retroviral vectors**. For the construction of the retrovirus vector-expressing mouse Pot1a, full-length cDNA of mouse Pot1a (OriGene Technologies, Inc.) was ligated into pMY-IRES-GFP (provided by Dr. Kitamura; University of Tokyo, Institute of Medical Science) or pMY-IRES-monomeric Kusabira-Orange1 (mKO1). For construction of Pot1a-shRNA retroviral vectors, oligonucleotides were inserted into pRetroU6-PGK/EGFP (gift from Dr. Takahiko Hara, The Tokyo Metropolitan Institute of Medical Science) at the BglII and HindIII sites and cloned as a retroviral vector.

**Retroviral transduction**. Freshly isolated LSK, LSKFlt3− cells or LSKCD41−CD48 −CD150+ cells were pre-cultured for 1 day in SF-O3 medium (EIDIA Co., Ltd.) in the presence of 0.1% BSA, 100 ng ml−1 stem cell factor (SCF) (PeproTech), and 100 ng ml−1 thrombopoietin (TPO) (PeproTech), and then transfected with retrovirus-expressing Pot1a or shPot1a on RetroNectin™ (Takara Bio Inc.) using Magnetofection™ (OZ Biosciences) according to the manufacturer's instructions. The cells were then cultured for one day at 37 °C in 5% CO₂. The efficiency of Pot1a overexpression or suppression in HSCs was confirmed by Q-PCR.

**Microarray analysis**. Aged LT-HSCs (60 week-old) were cultured in SF-O3 medium in the presence of BSA (0.1%), SCF (100 ng ml−1), and TPO (100 ng ml−1)

with control or MTM-POT1a protein for 3 days. After culture total RNA was isolated from cells using RNeasy Mini Kit (Qiagen) according to the manufacturer's instructions. RNA samples were quantified by an ND-1000 spectrophotometer (NanoDrop Technologies, Wilmington, DE) and the quality was confirmed with a 2200 TapeStation (Agilent technologies, Santa Clara, CA). The cRNA was amplified, labeled with 10 ng of total RNA using GeneChip® WT Pico Kit and hybridized to a Affymetrix GeneChip® Mouse Gene 2.0 ST Array array according to the manufacturer's instructions. All hybridized microarrays were scanned by an Affymetrix scanner. Relative hybridization intensities and background hybridization values were calculated using Affymetrix Expression Console™. Raw signal intensities for each probe were calculated from hybridization intensities. Raw signal intensities were log2-transformed and normalized by RMA and quantile algorithms[51] with Affymetrix® Expression Console™ 1.1 software. Gene set enrichment analysis (www.broadinstitute.org/gsea) was performed to determine whether the predefined gene sets were enriched in control or MTM-POT1a treated aged HSCs. The microarray data have been deposited in the Gene expression omnibus (GEO, http://www.ncbi.nlm.nih.gov/geo/) database, and have been assigned accession numbers GSE86386 and GSE85016.

**Production of MTM proteins**. The plasmid pET28a-MTM/N1-POT1a or POT1 was constructed by cloning full-length Pot1a or POT1 cDNA into the pET28a-MTM N1 vector (Supplementary Fig. 7). His-MTM-POT1a and POT1 were expressed in E. coli BL21 (DE3) cells transformed with pET28a-MTM/N1-POT1a or pET28a-MTM/N1-POT1 plasmids. Cells were grown at 37 °C in LB medium supplemented with kanamycin (100 μg ml−1) to reach an optical density (OD600 nm ~ 0.6). To induce the expression of the recombinant protein, 0.1 mM isopropyl thiogalactoside was added to the culture medium and the cells cultured for an additional 3 h at 37 °C. After culture, the cells were harvested. Recombinant protein was purified using a Ni-NTA purification system (Life Technologies Corporation), according to the manufacturer's instructions. The His-MTM peptide was used as the control. Mutant forms of Pot1a were constructed by cloning Pot1aΔOB or Pot1aΔTBD cDNA into the pET28a-MTM N1 vector. MTM-POT1aΔOB and MTM-POT1aΔTBD were purified by same procedure as described above.

**Long-term culture and colony formation assay**. To analyze the effect of the overexpression or knockdown of Pot1a on colony-forming activity, Pot1a- or shPot1a-transduced LSK cells were cultured for 1–4 weeks. After in vitro culture, GFP+ LSK cells were sorted and cultured in MethoCult™ GF M3434 medium for mouse cells and MethoCult™ H4034 for human cells (Stemcell Technologies).
    To analyze the effect of MTM-POT1a on the maintenance of HSC colony formation activity, LSKCD41−CD48−CD150+ cells were cultured for 10 days in SF-O3 medium in the presence of 0.1% BSA, 100 ng ml−1 SCF, and 100 ng ml−1 TPO with or without MTM-POT1a. Lin− cells were sorted and their ability to form colonies was then assessed. To analyze the effect of MTM-POT1a on the maintenance of HSC numbers, LT-HSCs (LSKCD41−CD48−CD150+ cells) were cultured for 3 weeks in SF-O3 medium in the presence of 0.1% BSA, 100 ng ml−1 SCF, and 100 ng ml−1 TPO with or without MTM-POT1a. After culture, the number of LSK cells and LT-HSCs was examined. For examination of colony formation in hCB HSCs after in vitro culture, Lin−CD34+CD38−CD45RA−CD90+ CD49f+ cells were cultured for 10 days in StemSpan SFEM (Stemcell Technologies) in the presence of 100 ng ml−1 SCF, 100 ng ml−1 rhFL, and 20 ng ml−1 TPO with or without MTM-POT1. After culture, Lin− cells were sorted and cultured in MethoCult™ H4034 medium.

**In vitro culture and immunostaining**. To analyze the effect of MTM proteins (control, Pot1a, Pot1aΔOB, and Pot1aΔTBD) on the inhibition of DDR in HSCs, LSKCD41−CD48−CD150+ cells were cultured for 10 days. LT-HSCs were then re-sorted from cultures and stained with either anti-TRF1 as a marker of the telomeric DNA region, anti-53BP1 as a marker of DNA damage, or anti-RPA32 and -phosphor-Chk1 for ATR signaling.

**BMT assay of Pot1a-overexpressed and knockdown HSCs**. To evaluate the function of Pot1a in the regulation of LTR activity, Ly5.1+ LSK cells (8 week-old) were transduced with retroviruses expressing Pot1a or Pot1a shRNAs. After retroviral transduction, GFP+LSK cells (5 × 10^3 cells per mice) were transplanted into lethally irradiated Ly5.2+ mice using 2 × 10^5 competitor cells (first BMT). The percentage of donor-derived GFP+ cells in PB was analyzed monthly by flow cytometry. After 16 weeks of the first BMT, the percentages of donor-derived (GFP+ Ly5.1+) cells in the BM LSK and LSKCD41−CD48−CD150+ fractions were analyzed. At the same time, 3 × 10^3 GFP+Ly5.1+ LSK cells were isolated from primary recipient mice and transplanted into lethally irradiated recipient mice (secondary BMT). The percentage of donor-derived GFP+ cells in PB was also analyzed monthly. After 16 weeks of the second BMT, the percentages of donor-derived cells in BM HSC fractions were analyzed in the same manner as the first BMT.
    To evaluate the effect of the overexpression of mutant forms of Pot1a, LT-HSCs (8 week-old) were transduced with retrovirus expressing Pot1a, Pot1aΔOB, or Po1aΔTBD and transplanted into lethally irradiated recipient mice. After 5 month of BMT, the percentages of GFP+ donor-derived cells were analyzed.

To analyze the function of Pot1a overexpression in the maintenance of HSC self-renewal in culture, Pot1a or control GFP-transduced Ly5.1[+] LSK cells (8 week-old) were cultured for 1–2 weeks in SF-O3 medium in the presence of 0.1% BSA, 100 ng ml[−1] SCF, and 100 ng ml[−1] TPO. After culture, GFP[+] LSK cells were transplanted into lethally irradiated recipient mice. Four months after BMT, the frequency of donor-derived GFP[+] cells in PB was analyzed by flow cytometry.

**Evaluation of LTR activity of HSCs treated with MTM-POT1a.** LSKCD41[−]CD48[−]CD150[+] cells (isolated from Ly5.1 mice) were cultured for 10 days in SF-O3 medium in the presence of 0.1% BSA, 100 ng ml[−1] SCF, and 100 ng ml[−1] TPO, with MTM-POT1a (350 ng ml[−1]) or control MTM protein. The cultures were divided by transferring one-half of the cultured cells into fresh medium at day 7. After 10 days of culture, cells were collected and transplanted into lethally irradiated Ly5.2 recipient mice (first BMT). The percentage of donor-derived cells in PB was analyzed monthly. Sixteen weeks post-BMT, the percentages of donor-derived B, T, and GM cells in PB, and in BMMNC, LSK, and LSKCD41[−]CD48[−]CD150[+] fractions were analyzed. For the second BMT, BMMNCs ($4 \times 10^6$ cells mouse[−1]) were isolated from primary recipient mice and transplanted into lethally irradiated recipient mice. The percentage of donor-derived GFP[+] cells in PB was again analyzed monthly. Sixteen weeks after BMT, the percentages of donor-derived B, T, and GM cells in PB, and in BMMNCs, LSK, and LSKCD41[−]CD48[−]CD150[+] fractions were again analyzed.

To evaluate the effect of mutant forms of Pot1a proteins on LTR activity of HSCs, LT-HSCs were cultured with MTM-(wild type) Pot1a, MTM-POT1aΔOB, or MTM-Po1aΔTBD for 10 days and transplanted into lethally irradiated recipient mice. After 4 months of BMT, the percentage of donor-derived (Ly5.1[+]) cells in PB was analyzed.

For the limiting dilution competitive BMT assay, LSKCD41[−]CD48[−]CD150[+] cells (from Ly5.1[+] mice) were cultured for 10 days in SF-O3 medium in the presence of 0.1% BSA, 100 ng ml[−1] SCF, and 100 ng ml[−1] TPO, with MTM-POT1a or control MTM. After culture, LSKCD41[−]CD48[−]CD150[+] cells were sorted and transplanted (20 or 50 cells per mice) into lethally irradiated Ly5.2 recipient mice. PB chimerism of >1.0% at 16 weeks after BMT was taken to verify long-term engraftment.

For the limiting dilution BMT assay of cultured human HSCs, hCB LT-HSCs (Lin[−]CD34[+]CD38[−]CD90[+]CD45RA[−]CD49f[+] cells) were cultured for 9 days with MTM-POT1 or control MTM. After the culture, $5 \times 10^3$ or $2 \times 10^4$ CD34[+] cells were transplanted into irradiated (2.4 Gy) NOG mice.

**Statistical analysis.** Significant differences between groups were determined using two-tailed Student's *t*-tests. Tukey's multiple comparison tests were used for multiple group comparisons. Significance of colocalization was assessed as described above. Frequency of LT-HSCs and statistical significance were determined using the ELDA software (http://bioinf.wehi.edu.au/software/elda/).

**Data availability.** The authors declare that all data supporting the findings of this study are available within the article or from the corresponding author upon reasonable request. Microarray data have been deposited in the GEO database under accession codes: GSE86386 and GSE85016.

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

## Acknowledgements

This work was supported by the funding program for Next Generation World-Leading Researchers (NEXT Program), a Scientific Research (B) (General), a grant-in-aid for Young Scientists (A) and a Challenging Exploratory Research from the Ministry of Education, Culture, Sports, Science and Technology (MEXT) of Japan, Tokyo Biochemical Research funding, a research grant from the Astellas Foundation for research on metabolic disorders, The Sumitomo Foundation, SENSHIN Medical Research Foundation, Daiichi Sankyo Foundation of Life Science and the European Union's Seventh Framework Programme (FP7/2007–2013) under grant agreement no: 306240 (SyStemAge).

## Author contributions

K.H. and F.A. designed and performed experiments, and analyzed and interpreted data; Y.M.I. and H.T. participated in performing experiments; B.D.M. analyzed the data; Y.M. and S.H. provided the pET28a-MTM vector and helped to prepare the MTM protein; K.H., B.D.M., T.S., and F.A. wrote the paper

## Additional information

**Competing interests:** The authors declare no competing financial interests.

