## [Peer Review File · Nature Communications]

Reviewers' Comments:

Reviewer #1 (Remarks to the Author)

The major claims of this paper are that Pot1a maintains HSC activity by preserving the stability of telomeric DNA and by preventing the production of reactive oxygen species (ROS). Due to these protective functions, the authors report that treatment with exogenous Pot1a maintains HSC self-renewal and function *ex vivo* and rejuvenates the activity of aged HSCs.

These are interesting and novel observations. The data are statistically compared in nearly all cases. While this reviewer is satisfied that these claims can be supported by the robust *in vitro* and *in vivo* data presented, the weakness of the paper is that the mechanisms through which this protein stabilizes the DNA and specifically how this protein modulates ROS production are not addressed. Nonetheless, this is a novel observation for the field of HSC biology.

Reviewer #2 (Remarks to the Author)

The manuscript by Hosokawa et al. focuses on investigating the role of Pot1a (Protection of telomeres 1a) in HSCs. The authors show that Pot1a levels are higher in HSCs and ST-HSCs compared to more differentiated hematopoietic cells and that the expression of Pot1a declines with aging. Additionally they show with knockdown and overexpression experiments that Pot1a, through modulation of the DNA damage response (DDR), can influence HSCs self-renewal. The authors demonstrate also that treatment with Pot1a improves activity of aged HSCs, and it results in enhanced maintenance of human HSCs *ex vivo*. Mechanistically the authors suggest that Pot1a reduces DNA damage at telomeres and decreases ROS. They imply that the overall beneficial effects of Pot1a in inducing HSC expansion are linked to protection from DNA damage but for example not linked to changes in proliferation etc.

The fact that Pot1a is specifically expressed in HSCs and it decreases upon aging was so far not reported. Also that Pot1a treatment can be used to enhance HSC activity upon aging or *ex vivo* expansion is an interesting and novel observation. A link between DNA damage and changes in HSC activity upon aging or stress is not novel. A role for Pot1a in DNA damage is also not novel. The topic of the manuscript is therefore of general interest. The data are overall well presented. However, some of the experiments are not clearly interpreted/explained, raising specific questions and over-interpretations that need to be addressed.

Major points:

1. The authors claim that loss of Pot1a compromises HSC activity (supp. Fig. 2). Data on Pot1a knockdown effects on HSC cell cycle and apoptosis would help supporting their conclusions.
2. In fig. 2g-i the authors show that Pot1a overexpression enhances HSC function *in vivo*. It is unclear if they cultured or not the cells before transplant. The increase in HSCs after the 1st BMT can be quantified based on the initial input of HSCs? Can be expressed as frequency over BM cells? Are ST-HSCs also affected? What about lineage contribution in the first BMT? Does cell cycle profile/apoptosis of HSCs change after Pot1a overexpression at the end of the first BMT? Upon secondary BMT the control show very low engraftment in all compartments. Is this the same with different control vectors? Do untransduced GFP- LSKs behave significantly different?
3. In fig. 3 the authors quantify TIFs. From the images it is difficult though to see the co-localization of 53BP1 and TRF1. How do the authors quantify the co-localization? Also, as observed by the authors (lines 158-159) the differences in TIFs frequencies largely depend on 53BP1. Do 53BP1 level change in Pot1a overexpression and knockdown settings? Is quantification by flow cytometry or western blot possible?
4. Fig. 4c-e shows reduction of ROS after Pot1a overexpression. It is unclear though why ROS levels are measured after 11 days in culture. GSEA analysis shows that the expression of reactive oxygen species metabolic process is already reduced at day 4. Additionally the authors don't show any link between ROS and DNA damage and I would suggest rephrasing the sentence in lines 174-

175.

5. In fig. 5b the authors show that Pot1a treatment inhibits apoptosis at week 1. The anti-apoptotic effect is though measured only in the context of Pot1a knockdown. In fig. 5c they show decreased TIFs in Pot1a treated cells. Finally in fig. 5d they show an increased number of HSCs in the Pot1a treated samples at week 3. According to the data on apoptosis at week 1, there is no difference between control and Pot1a treated HSCs, while HSC number in control decreases significantly. To fully support conclusions as in lines 195-196, data on HSC apoptosis, cell number and cycling should be provided (better at least two time points with some time consistency across measurements). The data on TIFs are not enough to conclude on DNA damage. I would also suggest to plot HSC number across time from beginning to end of treatment.

6. Line 199-200: the authors claim that treatment with Pot1a expands HSCs in vitro. Data in fig. 5 do not support this conclusion. Which data do the authors refer to?

7. Fig. 6a: why after 10 days of culture? How is the level of Pot1a in HSCs after the first BMT? What about TIFs after the first BMT?

8. Lines 227-231: TIFs staining is not enough to conclude on DNA damage. I would suggest rephrasing the sentences.

9. Fig. 7: Amount of Pot1a for treatment of young and aged cells was the same? What about Pot1a levels before BMT? Effect of Pot1a on cell cycle and apoptosis might be different on aged HSCs and to interpret the data it is critical to check for these parameters. How long were young and aged HSCs treated? Additionally, upon primary BMT transplant aged HSCs are known to expand. The authors detected a dramatic decrease. This aspect deserves to be discussed.

10. Fig 8f-g: data would be easier to interpret if number of cells during the culturing period would be plotted over time. Is the effect of Pot1a treatment dose-dependent?

11. The title of the manuscript and lines 275-276 link Pot1a to stress. Nevertheless the data show no real "stress" experiment. I would suggest slightly changing the title and this sentence (or providing more data with respect to stress: repetitive pIC treatment, 5FU, irradiation to support the statement).

Response to reviewers' comments

We thank reviewers #1 and #2 for their valuable comments and suggestions for improvements to our paper. We were pleased that both of the reviewers commented on the importance of our results and both of them offered constructive feedback on how to improve the paper, for which we are grateful. We have conducted extensive further experiments in response to these comments, as summarized below. We believe that the manuscript is substantially improved as a result.

Summary of major changes

In response to the reviewers' comments we have conducted the following additional experiments and analyses:

1. In our previous manuscript, we demonstrated that Pot1a maintains haematopoietic stem cell (HSCs) activity by two distinct mechanisms: (i) by preventing telomeric DDR and (ii) by a novel non-telomeric role in the inhibition of reactive oxygen (ROS) production. In the revised manuscript, we have gained further detail on the non-telomeric function of Pot1 in HSC maintenance. In particular, we show that Pot1a has a role in negatively regulating the mTOR pathway in HSCs. To do so we analyzed the gene expression patterns of HSCs after overexpression or knockdown of Pot1a. We found that overexpression of Pot1a in reduced the expression of *Mtor* and *Rptor* in both 8 and 90 week-old long-term (LT-) HSCs. Conversely, the knockdown of Pot1a in LT-HSCs increased the expression of *Mtor* and *Rptor*. Since it has been reported that the activation of mTOR pathway is involved in the production of reactive oxygen species (ROS) in HSCs (Chen, C. et al. 2008; Qian, P. et al. 2016), these results suggest that Pot1a has a role in this process. Although it is still unclear whether Pot1a directly regulates mTOR expression, our new observation indicates that the inhibition of mTOR signaling by Pot1a may be an important part of the non-telomeric function of Pot1a (in response to reviewer #1 and #2).
2. We have investigated the function of Pot1a/POT1 in the proliferation and apoptosis of LT-HSCs (in response to reviewer #2) and found that treatment with MTM-Pot1a/POT1 increased the number LT-HSC fraction in culture while knockdown of Pot1a inhibited proliferation and increased the LT-HSC apoptosis in culture.
3. We examined whether Pot1a overexpression increased the number of LT-HSCs after bone marrow (BM) transplantation (BMT) and found that Pot1a overexpression

significantly increased the number of donor-derived LT-HSCs in recipient mice after five months of BMT compared to controls (in response to reviewer #2).

4. We have examined the effect of Pot1a overexpression on the cell cycle of donor HSCs after primary and secondary BMT. We observed that Pot1a overexpression maintained Ki67⁻ quiescent population in the donor-derived LSK fraction after the primary and secondary BMT compared with controls (in response to reviewer #2).
5. We examined the function of Pot1a in the regulation of mitochondrial ROS level in LT-HSCs in culture. In the previous version of the manuscript, we showed that the overexpression of Pot1a inhibited the total intracellular ROS level in LT-HSCs. In addition to this effect, we found that Pot1a overexpression also inhibited mitochondrial ROS in LT-HSCs.
6. To examine the changes of 53BP1 level in Pot1a overexpression and knockdown settings, we have assessed the expression of 53BP1 in LT-HSC fraction by flow cytometry. We found that the level of 53BP1 expression in LT-HSCs was increased by Pot1a knockdown and was decreased by Pot1a overexpression. These data suggest that Pot1a prevents not only telomeric DNA damage response (DDR) but also total DDR in LT-HSCs (in response to reviewer #2).
7. In agreement with the comment of reviewer #2, we have changed the title of the manuscript.

Point-by-point responses to reviewers' comments

Reviewer #1:

The major claims of this paper are that Pot1a maintains HSC activity by preserving the stability of telomeric DNA and by preventing the production of reactive oxygen species (ROS). Due to these protective functions, the authors report that treatment with exogenous Pot1a maintains HSC self-renewal and function ex vivo and rejuvenates the activity of aged HSCs.

These are interesting and novel observations. The data are statistically compared in nearly all cases. While this reviewer is satisfied that these claims can be supported by the robust in vitro and in vivo data presented, the weakness of the paper is that the mechanisms through which this protein stabilizes the DNA and specifically how this

protein modulates ROS production are not addressed. Nonetheless, this is a novel observation for the field of HSC biology.

We very much appreciate this referee's encouraging comments. On the other hand, we agree that the clarification of the mechanism regulating ROS production by Pot1a is an important point of this work. To address this issue, we have now analyzed the gene expression patterns of HSCs upon overexpression or knockdown of Pot1a. We now show that overexpression of Pot1a reduces the expression of *Mtor* and *Rptor* mRNA in both young and aged long-term (LT-) HSCs whereas knockdown of Pot1a in LT-HSCs increases the expression of *Mtor* and *Rptor*. It has been reported that the activation of mTOR pathway is involved in the production of reactive oxygen species (ROS) in HSCs (Chen, C. et al. 2008; Qian, P. et al. 2016). Therefore, these results suggest that Pot1a prevents the production of ROS in HSCs via regulating mTOR activity and that the mTOR signaling pathway is a possible mediator of the non-telomeric function of Pot1a. We have added this data to the revised manuscript (new Fig. 4 and new Supplementary Fig. 10) and modified the text accordingly.

Reviewer #2:

The fact that Pot1a is specifically expressed in HSCs and it decreases upon aging was so far not reported. Also that Pot1a treatment can be used to enhance HSC activity upon aging or ex vivo expansion is an interesting and novel observation. A link between DNA damage and changes in HSC activity upon aging or stress is not novel. A role for Pot1a in DNA damage is also not novel.

The topic of the manuscript is therefore of general interest.

We thank this referee for the positive feedback.

Q1. The authors claim that loss of Pot1a compromises HSC activity (supp. Fig. 2). Data on Pot1a knockdown effects on HSC cell cycle and apoptosis would help supporting their conclusions.

A1. According to the reviewer's comment, we have now analyzed the effect of Pot1a knockdown on proliferation and apoptosis of LT-HSCs in culture. We found that knockdown of

Pot1a inhibited the proliferation of LT-HSC in culture whereas the total cell number was increased by Pot1a knockdown on day 7 and 11 of culture. We also found that Pot1a knockdown induced the apoptosis of LT-HSCs in culture. These findings suggest that Pot1a knockdown inhibits proliferation of LT-HSC by inducing the apoptosis. We have added this data to the revised manuscript (new Supplementary Fig. 2e-g) and modified the text accordingly.

Q2. (1) In fig. 2g-i the authors show that Pot1a overexpression enhances HSC function in vivo. It is unclear if they cultured or not the cells before transplant. (2) The increase in HSCs after the 1st BMT can be quantified based on the initial input of HSCs? Can be expressed as frequency over BM cells? Are ST-HSCs also affected? (3) What about lineage contribution in the first BMT? (4) Does cell cycle profile/apoptosis of HSCs change after Pot1a overexpression at the end of the first BMT? (5) Upon secondary BMT the control show very low engraftment in all compartments. Is this the same with different control vectors? Do untransduced GFP- LSKs behave significantly different?

A2-1. Concerning the experimental procedure of BMT shown in Fig. 2, we transduced the retrovirus expressing control-GFP or Pot1a (-IRES-GFP) into LSK cells. 3 days after of viral transduction, GFP-positive LSK cells were isolated and transplanted into lethally irradiated mice. To make the method clear, we have now modified the figure legend.

A2-2. As suggested by this reviewer, we have now analyzed whether Pot1a-overexpression increases the number of donor ST- and LT-HSCs after the 1st BMT. We found that Pot1a overexpression increased the frequency of donor-derived ST- and LT-HSCs was increased in whole BM (new Supplementary Fig. 5a) and Pot1a overexpression significantly increased the number of donor-derived LT-HSCs in recipient mice after five months of BMT compared with the control (new Supplementary Fig. 5b).

A2-3. In the revised manuscript, we have added the lineage contribution of Pot1a or control GFP-transduced donor HSCs. In contrast to Pot1a knockdown (Supplementary Fig. 2i), Pot1a overexpression increased the lymphoid cell fraction and decreased the myeloid cell fraction in donor cells compared with the control (this data is presented in new Fig. 2h).

A2-4. We analyzed the effect of Pot1a overexpression on the cell cycle of LT-HSC fraction after BMT. We found that Pot1a overexpression maintained Ki67⁻ quiescent population in the donor-derived LSK fraction after 1st and 2nd BMT (new Supplementary Fig. 5c), suggesting that the overexpression of Pot1a contributes to the maintenance of quiescent HSCs after BMT.

A2-5. As the reviewer suggested, control-GFP transduced cells showed low engraftment after 2nd BMT. In the results of the serial BMT assay shown in Fig. 2, we re-sorted GFP⁺Ly5.1⁺LSK cells after four months of 1st BMT and transplanted into secondary recipient mice (3000 cell/mice). We reason that this small cell number of input (donor cells) resulted in the low reconstitution of control-GFP transduced cells. We did not use a different control vector but we did multiple serial BMT assays and obtained the comparable results. We would like to emphasize that Pot1a-overexpressing cells showed higher engraftment in serial BMT even though a relatively small number of GFP⁺Ly5.1⁺LSK cells were transplanted as donor cells, indicating that Pot1a enhances HSC function. We did not test the engraftment capacity of retrovirus untransfected cells.

Q3. In fig. 3 the authors quantify TIFs. From the images it is difficult though to see the co-localization of 53BP1 and TRF1. How do the authors quantify the co-localization? Also, as observed by the authors (lines 158-159) the differences in TIFs frequencies largely depend on 53BP1. Do 53BP1 level change in Pot1a overexpression and knockdown settings? Is quantification by flow cytometry or western blot possible?

A3. Co-localization of 53BP1 and TRF1 was determined by image analysis as described by Dunn, K.W. et al. (Cell physiology, 2011) and Otsu N, et al. (Automatica, 1975). Briefly, subsequent to thresholding using Otsu's method, the extent of co-localization for each image was determined using Manders' co-localization coefficients

$$M1 = \frac{\sum_i G_{i, \text{colocal}}}{\sum_i G_i}$$

$$M2 = \frac{\sum_i R_{i, \text{colocal}}}{\sum_i R_i}$$

where R_i and G_i are the intensities of the i th pixel in the red and green channels respectively, and $R_{i, \text{colocal}} = R_i$ if $G_i > 0$ and zero otherwise (and similarly for $G_{i, \text{colocal}}$). Significance of co-localization was obtained by comparing observed co-localization coefficients with those obtained by independently scrambling the red and green channels 1×10^4 times. Note that since the number of red and green pixels may differ, M1 and M2 are not generally equal. Random scrambling of pixels does not preserve local spatial correlations in pixel intensities and can lead to overestimation of the significance of colocalization (Costes, S.V. et al. Biophys J, 2004). Therefore, the binary images from both channels were randomly divided into blocks

approximately the size of a TIF and these blocks (rather than individual pixels) were scrambled, as previously described by Costes, S.V. et al. (Biophys J, 2004). This method preserves local spatial correlations in pixel intensities and therefore provides a conservative null model. Significance of co-localization was determined by calculating the proportion of times that randomization increased the overlap coefficients M1 and M2 relative to the coefficients obtained prior to scrambling and by comparing observed and randomized co-localization coefficients using Welch's paired t-test. All analysis was restricted to regions identified as cell nuclei using co-staining for TOTO3 (blue).

Samples	STAINING		COLOCALIZATION OF GREEN CHANNEL WITH RED						COLOCALIZATION OF RED CHANNEL WITH GREEN					
	GREEN	RED	MEAN		SD		p-value	Welch's t-test p-value	MEAN		SD		p-value	Welch's t-test p-value
			observed	randomized	observed	randomized			observed	randomized				
8 week old HSCs	TRF1	53BP1	0.4661	0.3681	0.0525	0.0516	0.0153	<0.0001	0.5783	0.4533	0.048	0.0519	0.0102	<0.0001
90 week old HSCs	TRF1	53BP1	0.1787	0.1148	0.0306	0.0246	0.0063	<0.0001	0.6255	0.4137	0.0588	0.0635	0.0044	<0.0001
10 days cultured 8 week old HSCs/Control	TRF1	53BP1	0.7653	0.6533	0.0273	0.0318	0.0002	<0.0001	0.2244	0.1834	0.0156	0.0151	0.0002	<0.0001
10 days cultured 8 week old HSCs/MTM-Pot1a treated	TRF1	53BP1	0.6012	0.4609	0.0321	0.0337	<0.0001	<0.0001	0.2069	0.1454	0.0172	0.014	<0.0001	<0.0001
10 days cultured 90 week old HSCs/Control	TRF1	53BP1	0.5819	0.5078	0.029	0.0286	0.0064	<0.0001	0.2524	0.213	0.0168	0.0161	0.0045	<0.0001
10 days cultured 90 week old HSCs/MTM-Pot1a treated	TRF1	53BP1	0.7512	0.6388	0.0231	0.028	<0.0001	<0.0001	0.2442	0.2009	0.0136	0.0124	<0.0001	<0.0001
3 week cultured 8 week old HSCs/Control	TRF1	53BP1	0.4991	0.3198	0.0327	0.0321	<0.0001	<0.0001	0.4519	0.2893	0.0262	0.0265	<0.0001	<0.0001
3 week cultured 8 week old HSCs/MTM-Pot1a treated	TRF1	53BP1	0.5338	0.348	0.0405	0.041	<0.0001	<0.0001	0.3197	0.2071	0.0282	0.026	<0.0001	<0.0001
Control-shRNA transduced HSCs	TRF1	53BP1	0.2913	0.24	0.0578	0.0558	0.1653	<0.0001	0.4524	0.3741	0.0685	0.07	0.171	<0.0001
shPot1a transduced HSCs	TRF1	53BP1	0.2486	0.2009	0.0576	0.0532	0.1966	<0.0001	0.3444	0.2786	0.0671	0.0636	0.2014	<0.0001
Control GFP transduced HSCs/4M post BMT	TRF1	53BP1	0.5822	0.4229	0.1305	0.1353	0.1633	<0.0001	0.1326	0.0949	0.0442	0.0391	0.1785	<0.0001
Pot1a transduced HSCs/4M post BMT	TRF1	53BP1	0.6963	0.5256	0.1352	0.1525	0.1645	<0.0001	0.135	0.0951	0.0563	0.0426	0.1847	<0.0001
10 days cultured human CB HSCs/Control	TRF1	53BP1	0.509	0.4368	0.0358	0.0352	0.0092	<0.0001	0.4992	0.4229	0.0313	0.0313	0.0081	<0.0001
10 days cultured human CB HSCs/MTM-hPOT1 treated	TRF1	53BP1	0.4704	0.3174	0.0357	0.0342	<0.0001	<0.0001	0.3613	0.2495	0.0306	0.0281	0.0007	<0.0001
10 days cultured human CB HSCs/Control	TRF1	RPA32	0.3642	0.228	0.0322	0.0284	0.0001	<0.0001	0.3995	0.246	0.033	0.0287	<0.0001	<0.0001
10 days cultured human CB HSCs/MTM-hPOT1 treated	TRF1	RPA32	0.1494	0.1045	0.0217	0.018	0.0061	<0.0001	0.5428	0.4475	0.0552	0.0541	0.0906	<0.0001

[Figure caption] Statistics of co-localization of staining for TRF1 and 53BP1/RPA of the most of the images of this study. In total co-localization in >3750 cells from >1000 independent images were assessed.

We have added this information to the revised manuscript (Supplementary Information) and modified the text accordingly.

Although we observed significant colocalization of TRF1 and 53BP1 or RPA32 as shown above, we also found that the total number of 53BP1 foci was increased by Pot1a knockdown and was decreased by Pot1a overexpression. Moreover, according to the suggestion of this reviewer, we have now also examined the changes of 53BP1 level in Pot1a overexpression and knockdown by flow cytometry. We found that the total level of 53BP1 expression in LT-HSCs was increased by Pot1a knockdown and was decreased by Pot1a overexpression. These data suggest that Pot1a prevents not only telomeric DNA damage response (DDR) but also total DDR more generally in LT-HSCs. We have added this data to the revised manuscript (new Supplementary Fig. 6) and modified the text accordingly.

Q4. Fig. 4c-e shows reduction of ROS after Pot1a overexpression. It is unclear though

why ROS levels are measured after 11 days in culture. GSEA analysis shows that the expression of reactive oxygen species metabolic process is already reduced at day 4. Additionally the authors don't show any link between ROS and DNA damage and I would suggest rephrasing the sentence in lines 174-175.

A4. In the previous version of manuscript, we analyzed the ROS level in LT-HSCs on day 11 of culture to clarify the function of Pot1a under stressful conditions induced by long-term culture, while we did the microarray analysis after 4 days of culture (the day of sorting of GFP⁺ cells) to evaluate an early and direct effect of Pot1a overexpression. We appreciate that this was not clear and the text has been revised accordingly.

In the revised manuscript, we have also examined the function of Pot1a in the regulation of mitochondrial ROS level in LT-HSCs in culture. We transduced Pot1a into LT-HSCs and evaluated mitochondrial ROS by using MitoSOX™ reagent. We found that Pot1a overexpression inhibited mitochondrial ROS in LT-HSCs in culture. We have added this data to the revised manuscript (new Fig. 4c and f) and modified the text accordingly.

Furthermore, we also identified that overexpression of Pot1a inhibited the expression of *Mtor* and *Rptor* in both young and aged long-term (LT-) HSCs, while knockdown of shPot1a in LT-HSCs increased the expression of *Mtor* and *Rptor*. It has been reported that the activation of mTOR pathway is involved in the production of reactive oxygen species (ROS) in HSCs (Chen, C. et al. 2008; Qian, P. et al. 2016). Therefore, these results suggest that Pot1a prevents the production of ROS in HSCs via regulating mTOR activity. We have added this data to the revised manuscript (new Fig. 4 and new Supplementary Fig. 10) and modified the text accordingly (see also our response to reviewer #1).

Q5. In fig. 5b the authors show that Pot1a treatment inhibits apoptosis at week 1. The anti-apoptotic effect is though measured only in the context of Pot1a knockdown. In fig. 5c they show decreased TIFs in Pot1a treated cells. Finally in fig. 5d they show an increased number of HSCs in the Pot1a treated samples at week 3. According to the data on apoptosis at week 1, there is no difference between control and Pot1a treated HSCs, while HSC number in control decreases significantly. To fully support conclusions as in lines 195-196, data on HSC apoptosis, cell number and cycling should be provided

(better at least two time points with some time consistency across measurements). The data on TIFs are not enough to conclude on DNA damage. I would also suggest to plot HSC number across time from beginning to end of treatment.

A5. According to the comment from this reviewer, we have now analyzed the effect of Pot1a knockdown and MTM-Pot1a/POT1 treatment on cellular proliferation and apoptosis. We found that knockdown of Pot1a inhibited the proliferation and increased the apoptosis of LT-HSCs on day 7 and 11 of culture. Conversely, the treatment with MTM-Pot1a increased both young and aged LT-HSC fraction on days 4, 7, and 10 of culture. MTM-POT1 treatment also increased the number of human LT-HSCs on day 7 and 10 of culture. Both MTM-Pot1a and MTM-POT1 did not significantly affect the apoptosis of LT-HSC during culture. These data suggest that Pot1a prevents differentiation of LT-HSCs in culture. Similarly in cases when Pot1a expression has been lost (e.g. due to ageing) exogenous Pot1a improves HSC survival. We have added this data to the revised manuscript (new Fig. 5, 7, 8, and new Supplementary Fig. 2) and modified the text accordingly.

Q6. Line 199-200: the authors claim that treatment with Pot1a expands HSCs in vitro. Data in fig. 5 do not support this conclusion. Which data do the authors refer to?

A6. We agree with the reviewer that this section of the paper was not clear. To clarify further we have now compared the number of LT-HSCs before and after the 1st BMT in the revised manuscript. We find that Pot1a overexpression significantly increased the number of donor-derived LT-HSCs in recipient mice after five months of BMT compared with the control (new Supplementary Fig. 5b). We have revised the text accordingly.

Q7. Fig. 6a: why after 10 days of culture? How is the level of Pot1a in HSCs after the first BMT? What about TIFs after the first BMT?

A7. In Fig. 6, we did 10 days of culture to clarify whether MTM-Pot1a treatment maintains LT-HSC function during extended culture. Notably, the incorporated MTM-Pot1a level was significantly reduced in LT-HSCs 3 days after removal of MTM-Pot1a from the medium. Additionally, we could not detect remaining MTM-pot1a in the LT-HSCs after 1 month of BMT. These findings suggest that MTM-Pot1a treatment was effective only in the culture period.

Although we have not included this data in the manuscript for the reviewers' benefit we have included it below.

[Figure caption] (a) Schematic of the analysis of the intracellular level of MTM-Pot1a in HSCs. 8 week-old and 90 week-old LT-HSCs were cultured with MTM-Pot1a or control MTM protein for 10 days. After 10 days of culture, the medium was changed, and the culture was continued for the additional 3 days with or without MTM protein (b) Mean fluorescent intensity of MTM-Pot1a or control MTM protein on day 10 of culture. Data are expressed as the mean \pm SD (n = 6). (c) Mean fluorescent intensity of MTM-Pot1a or control MTM on day 13 of culture. Data are expressed as the mean \pm SD (n = 6).

(d) LT-HSCs were cultured with control or MTM-Pot1a for 10 days and transplanted into lethally irradiated recipient mice. After 1 month of BMT, intracellular MTM-protein levels were analyzed by FACS using anti-His Tag Ab. Data are expressed as the mean \pm SD (n = 5/group) (left panel). Representative FACS profiles are shown in the right panels.

Q8. Lines 227-231: TIFs staining is not enough to conclude on DNA damage. I would

suggest rephrasing the sentences.

A8. According to the reviewer's suggestion, we have modified text in the revised manuscript.

Q9. Fig. 7: Amount of Pot1a for treatment of young and aged cells was the same? What about Pot1a levels before BMT? Effect of Pot1a on cell cycle and apoptosis might be different on aged HSCs and to interpret the data it is critical to check for these parameters. How long were young and aged HSCs treated? Additionally, upon primary BMT transplant aged HSCs are known to expand. The authors detected a dramatic decrease. This aspect deserves to be discussed.

A9. We used the same amount of MTM-Pot1a for the treatment of young and aged HSCs. As described in response to the comment Q5, the treatment with MTM-Pot1a increased both young and aged LT-HSC fractions on days 4, 7, and 10 of culture but did not affect the apoptosis of LT-HSC during the culture period. Also, as described in response to the comment Q7, we could not detect remaining MTM-Pot1a in the LT-HSCs after 1 month of BMT.

This reviewer mentioned that aged HSCs were known to expand in numbers upon primary BMT. Because aged HSCs were cultured for 10 days before BMT, we speculate that the function of aged HSCs had declined during the culture. We have revised the text to make this clearer.

Q10. Fig 8f-g: data would be easier to interpret if number of cells during the culturing period would be plotted over time. Is the effect of Pot1a treatment dose-dependent?

A10. According to the reviewer's suggestion, we have examined the number of human LT-HSCs on days 7 and 10 of culture. We found that MTM-POT1 (50 and 500 ng/ml) treatment increased the number of human LT-HSCs on day 7 and 10 of culture, but we did not see a clear dose-dependent effect. Moreover, similar to MTM-Pot1a, MTM-POT1 did not affect the apoptosis of human LT-HSC during culture. We have added this data to the revised manuscript (new Fig. 8e) and modified the text accordingly.

Q11. The title of the manuscript and lines 275-276 link Pot1a to stress. Nevertheless the data show no real "stress" experiment. I would suggest slightly changing the title and this sentence (or providing more data with respect to stress: repetitive pIC treatment,

5FU, irradiation to support the statement).

A11. In agreement with the comment of reviewer #2, we have changed the title of the manuscript.

Reviewers' Comments:

Reviewer #1:

Remarks to the Author:

The authors have provided substantial additional data to support their claims and to find a pathway through which Pot1 can modulate the LT-HSC pool in a non-telomeric fashion. This is a substantially improved paper and addresses my prior concerns. This is an interesting and novel contribution to the literature and may be of significance (if translatable) in modulating LT-HSC activity in "older" subjects.

Reviewer #2:

Remarks to the Author:

The manuscript by Hosokawa et al. focuses on investigating the role of Pot1a (Protection of telomeres 1a) in HSCs.

In their revised version, they addressed most of the major points raised by the reviewers.

The also add novel data with respect to expression of changes in the mTOR pathways in with respect to changes in expression of Pot1a.

There are a couple of concerns remaining that need be addressed.

While the novel data implies are role of mTOR signaling, it does not establish any type of causality, and it still not clear which of all the pathways that the authors list as being correlatively linked to the phenotype are causatively involved in the phenotype. So either clearly establish causality with respect to a pathways, are please ensure that the text clearly states that these are strong, but still correlative data.

For example the novel title and parts of the abstract are misleading. The data presented do not support that conclusion "by protecting against DNA damage and metabolic alterations", same for the abstract (22.23). To imply causality is somewhat surprising as the abstract clearly mentions (27/28) that the changes are associated with, not more, but also not less.

A general concern of this reviewer remains with respect to the long-term cultivation of the cells with for example the Pot1 protein and implications of improvement of function for both young and aged HSCs. The comparison is to cells not treated with the protein, and not to non-treated cells. So the wording/conclusions should reflect that clearly, as some statements might imply that treated cells might perform better compared to non-cultivated, non-treated cells. Phenotypes linked to aging of HSCs are not only linked to changes in overall reconstitution and contribution to the B-cells compartment compared to the myeloid compartment. So re-activation of aged function upon ex vivo cultivation might be for example a more correct term for describing the role of exogenous Pot1a on aged HSCs.

Response to reviewers' comments

We appreciate the time and effort expended by the reviewers in reviewing the manuscript, as well as their valuable comments. We revised the text according to the reviewers' comments. We believe that the manuscript has been substantially improved.

Point-by-point responses to reviewers' comments

Reviewer #1:

We are glad to see that this reviewer acknowledged that the manuscript addressed the reviewers' concerns.

Reviewer #2:

1. In their revised version, they addressed most of the major points raised by the reviewers.

The also add novel data with respect to expression of changes in the mTOR pathways in with respect to changes in expression of Pot1a.

We thank this reviewer for the positive feedback.

2. While the novel data implies are role of mTOR signaling, it does not establish any type of causality, and it still not clear which of all the pathways that the authors list as being correlatively linked to the phenotype are causatively involved in the phenotype. So either clearly establish causality with respect to a pathways, are please ensure that the text clearly states that these are strong, but still correlative data.

We agree that our results, especially the role of mTOR signaling, was not sufficient to fully dissect the mechanisms by which Pot1a regulates HSC activity. To clarify this point we have now modified the text (particularly the abstract, results, and discussion sections) to make our conclusions more appropriate and consistent with the data.

3. For example the novel title and parts of the abstract are misleading. The data presented do not support that conclusion "by protecting against DNA damage and metabolic alterations", same for the abstract (22.23). To imply causality is somewhat surprising as the abstract clearly mentions (27/28) that the changes are associated with,

not more, but also not less.

We agree with this comment and, we have changed the title of the manuscript and modified the abstract appropriately.

4. A general concern of this reviewer remains with respect to the long-term cultivation of the cells with for example the Pot1 protein and implications of improvement of function for both young and aged HSCs. The comparison is to cells not treated with the protein, and not to non-treated cells. So the wording/conclusions should reflect that clearly, as some statements might imply that treated cells might perform better compared to non-cultivated, non-treated cells. Phenotypes linked to aging of HSCs are not only linked to changes in overall reconstitution and contribution to the B-cells compartment compared to the myeloid compartment. So re-activation of aged function upon ex vivo cultivation might be for example a more correct term for describing the role of exogenous Pot1a on aged HSCs.

We appreciate that our description of the control samples used in the long-term culture experiments was not clear. The text has been revised accordingly (result and Figure legend of Figure 7). We have also directed the reader to schematics shown in the Figures 4 and 7, and Supplementary Figures 2, 4, and 10 that we hope will make our experimental design transparent.

Reviewers' Comments:

Reviewer #2:

Remarks to the Author:

The authors addressed most of the remaining concerns of this reviewer.
It is now ready for publication.